# Optimization on multifractal loss landscapes explains a diverse range of geometrical and dynamical properties of deep learning

Andrew Ly & Pulin Gong ✉

Gradient descent and its variants are foundational in solving optimization problems across many disciplines. In deep learning, these optimizers demonstrate a remarkable ability to dynamically navigate complex loss landscapes, ultimately converging to solutions that generalize well. To elucidate the mechanism underlying this ability, we introduce a theoretical framework that models the complexities of loss landscapes as multifractal. Our model unifies and explains a broad range of realistic geometrical signatures of loss landscapes, including clustered degenerate minima, multiscale structure, and rich optimization dynamics in deep neural networks, such as the edge of stability, non-stationary anomalous diffusion, and the extended edge of chaos without requiring fine-tuning parameters. We further develop a fractional diffusion theory to illustrate how these optimization dynamics, coupled with multifractal structure, effectively guide optimizers toward smooth solution spaces housing flatter minima, thus enhancing generalization. Our findings suggest that the complexities of loss landscapes do not hinder optimization; rather, they facilitate the process. This perspective not only has important implications for understanding deep learning but also extends potential applicability to other disciplines where optimization unfolds on complex landscapes.

Deep learning has demonstrated remarkable achievements across diverse domains over the past decade[1], including image recognition[2], natural language processing[3], and even the longstanding protein folding problem[4]. A key element contributing to the effectiveness of deep learning is its optimizers, often rooted in simple iterative gradient descent methods[5]. These optimizers exhibit an extraordinary ability to dynamically navigate through complex loss landscapes, avoiding entrapment in local minima and ultimately converging to solutions that generalize well to previously unseen data. The surprising success of simple gradient descent-based optimizers in finding generalizing solutions raises the fundamental question of how such dynamic processes emerge in deep learning. Addressing this optimizability puzzle is crucial for advancing our fundamental understanding of deep learning, offering insights into the development of robust models and powerful optimization algorithms.

A prevalent theoretical framework for understanding optimization processes in machine learning draws extensively on principles from statistical mechanics[6–17]. In particular, since the early studies on learning from examples, Langevin dynamics has been employed as a theoretical approach to analyze optimization algorithms and to calculate typical learning curves[6,10–13]. More recently, Langevin-like models have been utilized to describe gradient descent-based optimizers in investigations of generalization in deep learning[18–21]. From a physical perspective, optimizers under this framework experience normal Brownian motion-driven diffusion. However, mounting empirical evidence challenges this understanding, revealing more intricate and diverse dynamics than Brownian motion featuring, for instance, non-stationary anomalous diffusive behaviors[22–24]. Correspondingly, models incorporating non-Gaussian heavy-tailed[21,25,26] and multiplicative

School of Physics, University of Sydney, Sydney, NSW, Australia. ✉e-mail: pulin.gong@sydney.edu.au

noise[27] have been considered, as these enhance the capacity for exploration of non-convex loss surfaces. According to random matrix theory, such dynamics with heavy-tailed steps may improve generalization by inducing a form of implicit regularization[15]. While this exploration of learning dynamics has primarily focused on the SGD algorithm, where noise arises from random mini-batching, evidence indicates that even full-batch gradient descent can exhibit stochastic-like chaotic dynamics[28].

In these cases, the absence of noise suggests that chaotic optimization dynamics must emerge from interactions with the geometric structure of the loss landscape. A significant discovery in this context is the edge of stability (EoS) phenomenon[29]. This phenomenon uncovers a surprising relationship between gradient descent dynamics and the underlying structure, with the leading Hessian eigenvalue along the gradient descent trajectory consistently hovering at or slightly above a stability threshold of 2/(learning rate). This observation challenges previous optimization theories based on the quadratic approximations of loss structure, where gradient descent is expected to diverge when the stability threshold is exceeded[30,31]. Consequently, the discovery of the EoS phenomenon emphasizes the need for a closer examination of the intricate interplay between gradient descent dynamics and the complexity of the loss landscape's geometric structure.

Empirical investigations into loss landscapes of deep learning have consistently demonstrated a strong correlation between the curvature of the landscape around a solution and generalization, measured through various curvature metrics[24,32–35]. While these curvature metrics are mostly local, studies adopting a statistical mechanics perspective typically quantify more global structures in the loss landscape[9,36–39]. Along this line, generalization has also been found to strongly correlate with the connectivity of solutions; the best generalization is associated with a rugged basin containing locally flat minima connected by highly degenerate low-loss paths[14]. Recent evidence also suggests that the loss landscape may exhibit multiscale features[28,40]. However, the geometric nature of the loss landscape that captures all these properties remains elusive. Furthermore, the fundamental questions of whether and how this geometric structure mechanistically relates to complex learning dynamics, including chaotic dynamics and the EoS, as well as generalization in deep learning, remain unclear.

To address these questions, we develop a theoretical framework that models loss landscapes as multifractal. Multifractality is characterized by a continuous (infinite) set of scaling exponents[41], and has been applied to understand the complexity of a range of systems, including the Anderson phase transition[42], turbulence[43], the anomalous scaling behavior of self-similar diffusion processes[44], the organization of internal representations in neural networks[9], heterogeneous neural network dynamics[45], and the non-equilibrium property of spin-glass systems as recently found[46]. We demonstrate that our model offers a coherent explanation that links a diverse set of geometrical signatures of loss landscapes—such as multiscale features, highly degenerate minima, low barriers, and clustering—with optimization dynamics, including the EoS phenomenon, extended edge of chaos, and non-stationary anomalous diffusion. This is significant because these phenomena have until now been treated in an isolated way with separate models for each phenomenon.

By adapting fractional calculus formalisms used for studying non-equilibrium physical systems[47], we further develop a fractional diffusion theory. This theory analytically illustrates how, by leveraging landscape-dependent annealing-like properties, the dynamics of gradient descent (GD) actively guide the optimizer toward large and smooth solution spaces that house well-connected flatter minima, thus benefiting generalization. Our results indicate that contrary to intuition, the complexity inherent in loss landscapes does not impede optimization; instead, it facilitates the process. This novel perspective holds potential applicability beyond machine learning, extending its relevance to diverse fields such as evolutionary biology and ecology, where optimization similarly unfolds on complex fitness landscapes[48,49].

## Results

### Multifractal loss landscape reproduces key properties found in realistic scenarios

We first introduce our model for deep neural network optimization. The capacity of a deep neural network $f : \mathbb{R}^{d_{in}} \to \mathbb{R}^{d_{out}}$ to fit the training data is measured by a loss function:

$$L(\boldsymbol{\theta}) = \frac{1}{N} \sum_{(\mathbf{x}, \mathbf{y}) \in \mathcal{D}} l(f(\mathbf{x}), \mathbf{y}), \tag{1}$$

where $\boldsymbol{\theta} \in \mathbb{R}^n$ denotes the parameters of the network, $\mathcal{D} = \{(\mathbf{x}_i, \mathbf{y}_i)\}_{i=1}^N$ is the training dataset of size $N$ with $\mathbf{x}_i \in \mathbb{R}^{d_{in}}$ and $\mathbf{y}_i \in \mathbb{R}^{d_{out}}$, and $l : \mathbb{R}^{d_{out}} \times \mathbb{R}^{d_{out}} \to \mathbb{R}$ is the single-sample loss. The training of a deep neural network can be formulated as the optimization problem of finding $\arg\min_{\boldsymbol{\theta}} L(\boldsymbol{\theta})$. Applying the GD algorithm to this problem, the network parameters evolve as:

$$\boldsymbol{\theta}_{t+1} = \boldsymbol{\theta}_t - \eta \nabla L(\boldsymbol{\theta}_t), \quad t = 0, 1, 2, \ldots \tag{2}$$

where $\eta$ is the learning rate and $t$ denotes the iteration.

To formulate the loss function $L$ as a multifractal landscape, we first introduce the pointwise Hölder exponent $H : \mathbb{R}^n \to (0, 1)$. Given a function $f : \mathbb{R}^n \to \mathbb{R}$, $H$ is defined as[50]:

$$H(\boldsymbol{\theta}) = \sup \left\{ \alpha : \lim_{\|\boldsymbol{\delta}\| \to 0} \frac{|f(\boldsymbol{\theta} + \boldsymbol{\delta}) - f(\boldsymbol{\theta})|}{\|\boldsymbol{\delta}\|^\alpha} = 0 \right\}, \tag{3}$$

where the supremum indicates the least upper bound of the set of $\alpha$ values for which the given limit becomes zero. Physically interpreting this equation, it indicates that the largest fluctuations of $f$ within a small displacement $\boldsymbol{\delta}$ from point $\boldsymbol{\theta}$ are of order $\|\boldsymbol{\delta}\|^{H(\boldsymbol{\theta})}$. Thus, $H(\boldsymbol{\theta})$ characterizes the roughness of $f$ at the point $\boldsymbol{\theta}$, with roughness increasing (i.e., larger fluctuations) as $H(\boldsymbol{\theta})$ decreases. Assuming a continuously differentiable pointwise Hölder exponent $H$ is prescribed, we can create a corresponding random Gaussian function $B_H$ with covariance[51,52]:

$$\langle B_H(\boldsymbol{\theta}_1) B_H(\boldsymbol{\theta}_2) \rangle \propto \|\boldsymbol{\theta}_1\|^{H(\boldsymbol{\theta}_1)+H(\boldsymbol{\theta}_2)} + \|\boldsymbol{\theta}_2\|^{H(\boldsymbol{\theta}_1)+H(\boldsymbol{\theta}_2)} - \|\boldsymbol{\theta}_1 - \boldsymbol{\theta}_2\|^{H(\boldsymbol{\theta}_1)+H(\boldsymbol{\theta}_2)}. \tag{4}$$

This covariance captures the spatial correlations in the structure of $B_H$ that give rise to fractal properties. Specifically, if the Hölder exponent $H$ varies across different points $\boldsymbol{\theta}$ (i.e., it is heterogeneous and non-uniform), we regard $B_H$ as multifractal. Otherwise, if $H$ is constant, $B_H$ is a self-similar monofractal with spatially uniform roughness (Fig. S1). Finally, we construct the loss landscape $L$ to be a coarse-grained realization of $B_H$ (see "Methods" for further details on constructing $L$ and see Supplementary Sec. 1 for additional information on $B_H$).

We proceed to elucidate the multifractal nature of the loss landscape $L$. This can be understood through two key observations: First, $B_H$ exhibits local asymptotic self-similarity. Essentially, this indicates that a dilation or scaling of $B_H$ about any point $\boldsymbol{\theta}$ increasingly resembles $B_{H(\boldsymbol{\theta})}$ as the scale factor grows (see Supplementary Sec. 1 for further details). Second, $B_{H(\boldsymbol{\theta})}$ represents a self-similar monofractal with a fractal dimension $n + 1 - H(\boldsymbol{\theta})$ which obeys the following scaling law: $a^{H(\boldsymbol{\theta})} B_{H(\boldsymbol{\theta})}(\boldsymbol{\theta}')$ is identically distributed as $B_{H(\boldsymbol{\theta})}(a\boldsymbol{\theta}')$ for a constant $a$. Consequently, the pointwise Hölder exponent governs local power-law scaling. In cases where the pointwise Hölder exponent is non-uniform, then a range of fractal scaling exponents coexist across $B_H$; the spatial variation of these exponents is controlled by the function $H$. Given coexisting fractal exponents is a characteristic of multifractality,

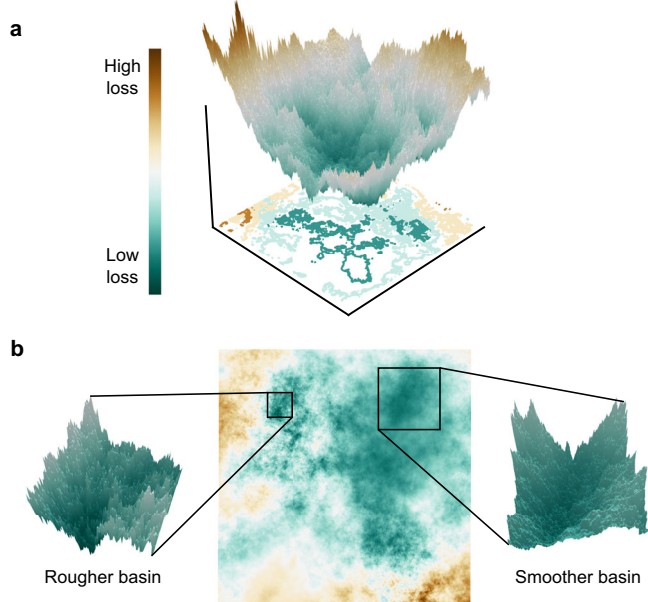

**Fig. 1 | A loss landscape with multifractal structure. a** A realization of a 2-D multifractal loss landscape $L$, with contours at every 20th percentile of loss shown on the $x$-$y$ plane. **b** Projection of the loss landscape in 2-D (center). To highlight its multiscale structure, we zoom in on the boxed regions, including a rougher basin with Hölder exponent $H \approx 0.3$ (left) and a smoother basin with $H \approx 0.7$ (right).

we regard $B_H$ as multifractal. Mathematically speaking, $B_H$ is only multifractal-like; however, for simplicity, we use the terms multifractal and multifractal-like interchangeably (see Supplementary Sec. 2 for further clarification on these technical details). Through its construction, the loss landscape $L$ inherits these multifractal properties at all scales down to the resolution of the coarse-graining.

We now construct a minimal realization (2-D in this instance) of $L$ (Fig. 1) to illustrate how the multifractal loss landscape reconciles many key properties found in real loss landscapes. Such a minimal modeling approach, grounded in the principles of statistical physics, is a common strategy in theoretical investigations of deep learning[6–9,17]; it enables analytical and computational tractability while providing valuable insights, as seen in previous studies[28,40,53,54]. In essence, the process of constructing $L$ involves the following steps: 1) specifying a continuously differentiable pointwise Hölder exponent $H : \mathbb{R}^n \rightarrow (0, 1)$, 2) generating a random realization of $B_H$, whose pointwise Hölder exponent is $H$ almost surely, and discretizing it on a Cartesian grid, and 3) linearly interpolating to calculate the gradient $\nabla L$ at an arbitrary point (see "Methods" for further details). Here, we set $0.3 \leq H(\boldsymbol{\theta}) \leq 0.7$ to demonstrate a relatively broad range of roughness. It is important to note that the properties we discuss regarding $L$ are generalizable; see Supplementary Sec. 11 for further clarification.

To validate the presence of multifractal features in the loss landscape, we perform multifractal analysis to quantify the diverse scaling properties across different regions of the landscape. This analysis involves calculating the multifractal singularity spectrum, defined as $f(\alpha) \equiv d_H(\{\boldsymbol{\theta}: H(\boldsymbol{\theta}) = \alpha\})$, where $d_H$ is the Hausdorff fractal dimension. Thus, $f(\alpha)$ represents the space-filling capacity of the set of points that have a pointwise Hölder exponent value of $\alpha$. We compute this spectrum using the wavelet leaders multifractal formalism[50]. As shown in Fig. 2a, the multifractal singularity spectrum $f(\alpha)$ reveals a continuous set of scaling exponents, which is a characteristic feature of multifractality. In contrast, a monofractal counterpart ($H = 0.5$ globally uniform) exhibits a trivial $f(\alpha)$ with point support solely at $\alpha = 0.5$. In terms of these local scaling properties, the loss landscape $L$ demonstrates a multiscale nature. To illustrate this directly, we zoom in on

two different regions of $L$ in Fig. 2b. Both regions, despite differences in roughness, display larger basins that host clusters of smaller minima.

We next analyze an empirically important property of the loss landscape $L$, its curvature. The total curvature at a point can be measured by $||\mathbf{H}||_F^2 = \sum_i \lambda_i^2$, where $|| \cdot ||_F$ denotes the Frobenius norm, $\mathbf{H}$ is the Hessian matrix, and $\lambda_i$ are the Hessian eigenvalues satisfying $|\lambda_1| \geq |\lambda_2| \geq \cdots \geq |\lambda_n|$; see Supplementary Sec. 3 for a comparison to other curvature metrics, showing that all metrics consistently identify flatter minima within the multifractal landscape (Fig. S3). In our model, we analytically derive for 2-D (see "Methods"):

$$\langle ||\mathbf{H}||_F^2 \rangle \propto r(H)\chi^{2H-2}, \tag{5}$$

where $r(H) = 16 - 2^{2H+2} + 2^{2H} - 2^{3H-1}$ is a coefficient relating to roughness (because it strictly decreases to zero as Hölder exponent $H$ increases to one, as shown in Fig. S10), and $\chi$ is the Cartesian grid spacing. As shown in Fig. 2b, we further validate this relation numerically which, additionally, reveals a wide range of curvature in our loss landscape spanning several orders of magnitude. Importantly, equation (5) states that the expected curvature increases with roughness. As a result, flatter minima, on average, reside in smoother basins. Empirical results consistently indicate that a decrease in $\langle ||\mathbf{H}||_F^2 \rangle$, along with other curvature metrics, correlates with improving generalization[33,35]. Thus, according to equation (5), the learning dynamics of GD on loss landscape $L$ should navigate towards smoother basins if it is to achieve flat, generalizing solutions; this will later be explained by our fractional diffusion theory.

Furthermore, clustered solutions in a smoother basin of the loss landscape $L$ are highly degenerate (i.e., have near-identical loss values) and are separated by low barriers. In other words, solutions in smoother basins are well-connected[14,39]. This property is illustrated by considering the cluster shown in the inset of Fig. 2c. The cluster can be uniformly bounded by a contour at $L = 0.5$, which is a tight upper bound of the set of loss values of minima in the cluster ([0.337, 0.498]); Fig. 2e shows a histogram of these loss values. This observation implies that the minima are separated by barriers whose heights are no larger than the range of loss values. Given that this range is narrow (compared to the global range of loss [0, 3.2]), we infer that the barriers are low and, simultaneously, the solutions are highly degenerate. Further analysis of the minimum Hessian eigenvalue in our multifractal model similarly suggests that local minima are confined to a low-loss band (Fig. S2). Taken together, the clustering of highly degenerate solutions within smoother basins aligns with theoretical and empirical characterizations of the global structure of loss landscapes from a statistical mechanics perspective[9,36–39]. A major implication of these studies is that the solution space could be described as a manifold of well-connected, non-isolated minima that is highly navigable for an optimizer. Importantly, such connectivity among solutions has been found to strongly correlate with generalization[14].

## Deep neural network examples

We now empirically relate the multifractal model to the loss landscape of realistic deep neural networks. In particular, by applying the filter normalization method in ref.[55], we visualize and characterize two-dimensional sections of the loss landscape centered on the location of the optimizer at different epochs during the training of a VGG-16 network on the CIFAR-10 dataset (see "Methods" for further details on the experimental configuration and other types of networks and data). The orientation of the two-dimensional sections is defined by filter-normalized random directions. This method is widely used because it enables meaningful comparisons of loss structure by preventing distortions caused by the scale invariance of networks with ReLU non-linearities[55–58]. Figure 3a shows the loss surface after 20 epochs. The surface exhibits a low-loss basin at the optimizer position that appears smoother than surrounding regions of higher loss. To quantify these

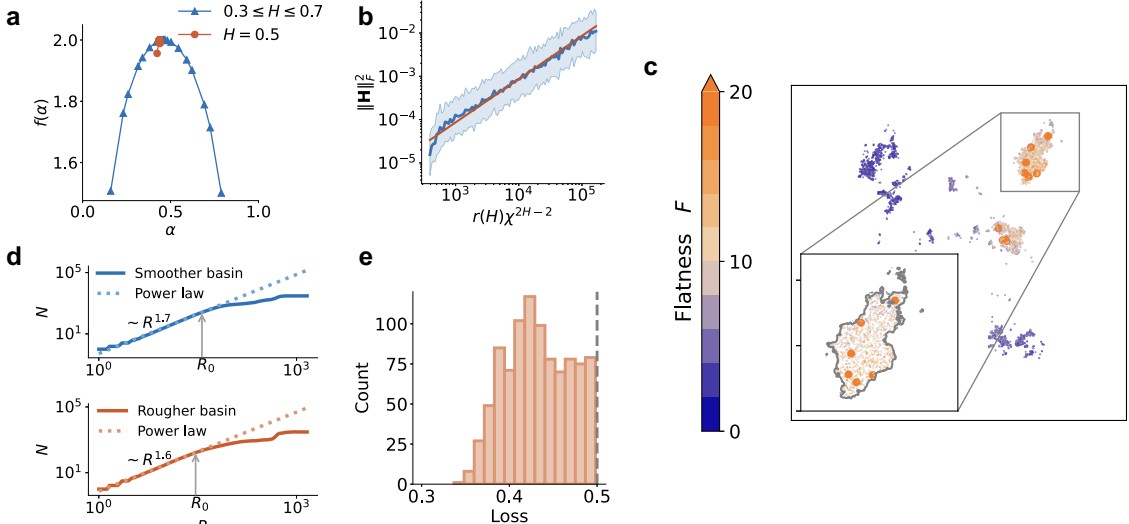

**Fig. 2 | Properties of the multifractal model. a** Multifractal singularity spectrum $f(\alpha)$ of loss landscape $L$. For comparison, $f(\alpha)$ is also calculated for a counterpart loss landscape constructed from $B_H$ with constant Hölder exponent $H = 0.5$. **b** Validation of equation (5) for 10,000 random points sampled from $L$. The blue curve indicates the empirical mean of $||\mathbf{H}||_F$ in 100 logarithmically spaced bins. The shaded region corresponds to 2 standard deviations of $\log_{10}||\mathbf{H}||_F$. The red line illustrates a proportional relationship between $||\mathbf{H}||_F^2$ and $r(H)\chi^{2H-2}$. **c** Dots represent the 3000 lowest-loss minima, with color indicating flatness ($F \equiv 1/\sqrt{||\mathbf{H}||_F}$). The ten flattest minima are depicted notably larger. The inset

offers a closer look at the smoother basin from Fig. 1b, showcasing a cluster of relatively flatter minima consistent with the correlation in equation (5). The cluster is uniformly bounded by a contour at $L = 0.5$ (gray curve). **d** Power-law growth of the number of low-loss minima $N$ within a Euclidean distance $R$ from a minimum, indicating clustering in both the smoother and rougher basins from Fig. 1b. The exponent $d_f$ of the power-law fits (dotted lines) reveals the fractal dimension (or space-filling capacity) of the cluster. **e** Histogram of loss values of minima within the cluster visualized in the inset of **c**. The dashed gray line indicates the contour level.

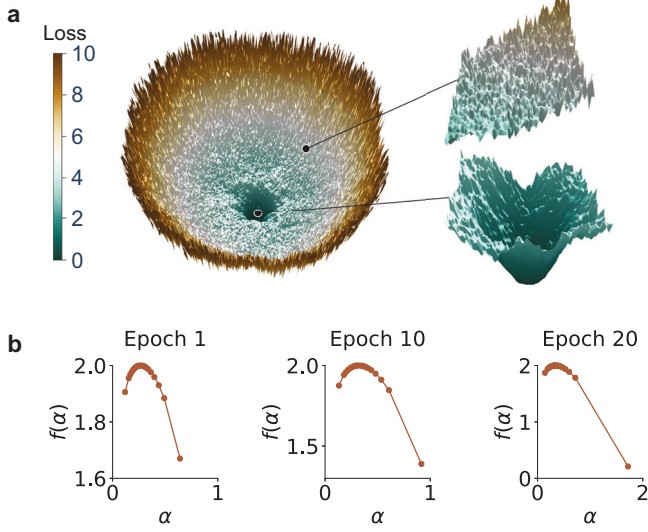

**Fig. 3 | Loss landscape of a deep neural network. a** Two-dimensional section of the loss landscape centered at the position of the optimizer after 20 epochs of training a VGG-16 network on the CIFAR-10 dataset. Loss values greater than 10 are clipped to avoid obscuring details at smaller loss values. To highlight multifractal structure, we zoom in on a rougher region (top) and a smoother basin (bottom). **b** The multifractal singularity spectrum $f(\alpha)$ of two-dimensional sections of the loss landscape at various training stages: after 1, 10, and 20 epochs.

variations in roughness across the surface, we calculate the multifractal singularity spectrum, which shows a continuous set of scaling exponents (Fig. 3b), indicating the presence of multifractal structure. In addition, we find that the singularity spectrum broadens towards larger exponents $\alpha$ during training. This is because the loss visualization is centered at the optimizer, and the optimizer navigates towards a smoother basin. This finding is further supported by estimations of

the pointwise Hölder exponent across the two-dimensional visualizations, which quantifies the precise spatial variation in roughness (see Supplementary Sec. 4). Further characterizations of roughness do not change significantly after 20 epochs, which approximately coincides with when accuracy reaches a maximum and plateaus. Correspondingly, as our fractional diffusion theory will later demonstrate, GD in a multifractal loss landscape exhibits a preference for smoothness that benefits generalization. In Supplementary Sec. 4, we show these results are robust to changes in training data and optimizer (Figs. S5–6), characterize fractal clustering in these surfaces as done in our model (Fig. S7), and demonstrate the general applicability of our analysis to real loss landscapes with multifractal structure (Figs. S8–9).

## Dynamics of GD optimization on the multifractal loss landscape

We next elucidate that gradient descent (GD) navigating through the multifractal loss landscape captures and explains a diverse set of deep learning dynamics. Additionally, we establish the robust explanatory capability of our theoretical framework by demonstrating its general applicability.

Contrary to previous wisdom for efficient optimization based on assumptions of $L$-smoothness regularity[30,31], recent studies on the EoS phenomenon have shown that larger learning rates $\eta$ marginally violating the stability condition (i.e., $\eta > 2/\lambda_1$, where $\lambda_1$ is the leading Hessian eigenvalue) lead to superior performance[29]. We note that in recent optimization theories with more realistic regularity assumptions, such as Lipschitz continuous and bounded loss[26], violating the stability condition does not preclude learning. Since the EoS phenomenon is a characteristic feature of neural network training with GD, we demonstrate here that it naturally arises in our model. To illustrate this, we systematically vary both the initialization and the learning rate $\eta$. We find that the leading Hessian eigenvalue $\lambda_1$ consistently reaches just above the stability threshold, $2/\eta$, regardless of the initialization, showing a hallmark of the EoS phenomenon called sharpness concentration[53]. This behavior also occurs in a wide range of learning rates, exhibiting a second hallmark called sharpness adaptivity.

Notably, we find that this wide range can be further increased by expanding the range of the pointwise Hölder exponent $H$ in the multifractal landscape. This indicates that multifractality is mechanistically related to an extended EoS (see Supplementary Sec. 10 and 11 for control studies demonstrating this).

We now demonstrate that GD operating in the regime of the EoS phenomenon exhibits chaotic dynamics, specifically at an extended edge of chaos. While a rigorous definition of chaos requires several properties, the most relevant one for network optimization is sensitivity to initial conditions, which we use here to characterize chaos. We quantify this sensitive dependence on initial conditions by comparing how an infinitesimally perturbed GD trajectory $\boldsymbol{\theta}_t + \delta\boldsymbol{\theta}_t$ evolves over time relative to the original trajectory $\boldsymbol{\theta}_t$. The divergence of the two trajectories after a single iteration can be measured by the local Lyapunov exponent, computed as:

$$\lambda(\boldsymbol{\theta}_t, \hat{\mathbf{n}}_t) \equiv \ln \frac{\| \delta\boldsymbol{\theta}_{t+1} \|}{\| \delta\boldsymbol{\theta}_t \|} = \frac{1}{2}\ln(\hat{\mathbf{n}}_t^T \mathbf{A}(\boldsymbol{\theta}_t)^T \mathbf{A}(\boldsymbol{\theta}_t)\hat{\mathbf{n}}_t), \qquad (6)$$

where $\hat{\mathbf{n}}_t \equiv \delta\boldsymbol{\theta}_t/\|\delta\boldsymbol{\theta}_t\|$ is a unit vector in the direction of the displacement $\delta\boldsymbol{\theta}_t$, and $\mathbf{A}(\boldsymbol{\theta}_t) \equiv \partial\boldsymbol{\theta}_{t+1}/\partial\boldsymbol{\theta}_t = \mathbf{I} - \eta\mathbf{H}(\boldsymbol{\theta}_t)$ is the stability matrix of the GD algorithm (see Supplementary Sec. 5 for its mathematical derivation). Given a perturbation in the direction of a Hessian eigenvector $\hat{\mathbf{n}}_t = \mathbf{v}_i$, with corresponding eigenvalue $\lambda_i$, the local Lyapunov exponent is $\lambda(\boldsymbol{\theta}_t, \mathbf{v}_i) = \frac{1}{2}\ln(1 - \eta\lambda_i)^2$. As a result, the displacement is:

$$\begin{cases} \text{expanding if } \lambda(\boldsymbol{\theta}_t, \mathbf{v}_i) > 0, \text{ i.e. }, \lambda_i < 0 \text{ or } \lambda_i > 2/\eta \\ \text{marginal if } \lambda(\boldsymbol{\theta}_t, \mathbf{v}_i) = 0, \text{ i.e. }, \lambda_i = 0 \text{ or } \lambda_i = 2/\eta \\ \text{contracting if } \lambda(\boldsymbol{\theta}_t, \mathbf{v}_i) < 0, \text{ i.e. }, 0 < \lambda_i < 2/\eta. \end{cases} \qquad (7)$$

Thus, GD operating above the stability threshold with $\eta > 2/\lambda_1$ is a sufficient condition for the emergence of chaos in the direction of the leading eigenvector $\mathbf{v}_1$ at single-iteration timescales. Such chaos can extend to longer timescales if the time-average of the local Lyapunov exponent is positive (see Supplementary Sec. 5 for the derivation):

$$\lambda \equiv \lim_{t \to \infty} \frac{1}{t}\ln\frac{\| \delta\boldsymbol{\theta}_t \|}{\| \delta\boldsymbol{\theta}_0 \|} = \lim_{t \to \infty} \frac{1}{t}\sum_{i=0}^{t-1}\lambda(\boldsymbol{\theta}_i, \hat{\mathbf{n}}_i), \qquad (8)$$

where $\lambda$ is the maximal Lyapunov exponent characterizing the long-term exponential divergence of nearby trajectories. Thus, we experimentally quantify chaos across long time intervals by using Sprott's algorithm[59] to estimate the maximal Lyapunov exponent (see "Methods" for implementation details). We find that the exponent $\lambda$ is generally positive once the learning rate $\eta > 2/\lambda_1$, implying that the chaotic dynamics indeed extends across long time intervals; across the range of learning rates exhibiting the EoS phenomenon, $\lambda$ fluctuates about a mean of 0.10 with a standard deviation of 0.23. Importantly, because the stability threshold coincides with the onset of chaos by equation (7), the EoS phenomenon in deep learning, wherein $\lambda_1$ hovers just above $2/\eta$, indicates that GD naturally operates near the edge of chaos.

The broad range of EoS phenomenon, which expands with the range of Hölder exponents, demonstrates that the edge of chaos exists across an extensive region without requiring parameter (e.g., learning rate) fine-tuning. The concept of the edge of chaos has long been regarded as an optimal condition for computation in physical systems[60] and has since been extensively applied to understand biological evolution[61] and computation in recurrent neural networks[62]. This result not only broadens the concept of the edge of chaos beyond a critical point that requires precise parameter fine-tuning, as demonstrated in these previous studies as well as in deep learning research[63,64], but also illustrates how this extended edge of chaos emerges from the perspective of multifractal landscapes.

We proceed to illustrate that the dynamics at the edge of chaos produce the stochastic-like behavior of deterministic GD, such as convergence in distribution. In particular, the instability of GD when $\eta > 2/\lambda_1$ fundamentally drives the optimization trajectory out of local minima. Yet, simultaneously, this trajectory can be uniformly bounded due to the large-scale geometrical structure of loss landscape $L$ (e.g., the black trajectory in Fig. 4b is ultimately bounded within the vicinity of the top-right basin). This interplay between local divergence (from local minima) and global confinement (within a basin containing a cluster of local minima) combines to yield unstable convergence, not towards a single fixed point, but in distribution across a solution space. A consequence of convergence in distribution is that the loss continually fluctuates at short timescales but decreases or remains constant at long timescales (see the blue curves and the green curve in Fig. 4c), demonstrating another key characteristic feature observed in the EoS phenomenon in deep learning[29].

Furthermore, the variance of the equilibrium distribution continuously increases with $\eta$ until large enough that the distribution floods over even the largest basin. In this extreme scenario, the loss along a trajectory exhibits excessively large fluctuations (green curve in Fig. 4c), which precludes convergence to a low-loss solution space and is thus impractical for network training. Conversely, GD exhibits stable convergence if the learning rate $\eta$ is sufficiently small; the loss decreases monotonically, but slowly, until it plateaus at a nearby local minimum (red curve in Fig. 4c). This small-$\eta$ behavior is another suboptimal scenario where training speeds are impractically slow[29].

We now characterize the diffusive dynamics of GD on multifractal loss landscapes by calculating the time-averaged mean squared displacement (TAMSD):

$$\overline{\delta^2}(t_w, \tau) \equiv \frac{1}{T}\sum_{t=t_w}^{t_w+T} \| \boldsymbol{\theta}_{t+\tau} - \boldsymbol{\theta}_t \|^2, \qquad (9)$$

where $\overline{\delta^2}$ measures the squared displacement of the optimizer after a lag time $\tau$ with respect to a reference position at the waiting time $t_w$, time-averaged with a window of size $T$. A linear TAMSD where $\overline{\delta^2} \propto \tau^\alpha$ with diffusion exponent $\alpha = 1$ represents normal diffusion, e.g., Brownian motion. Deviation from linearity is termed anomalous diffusion. In particular, $1 < \alpha < 2$ is super-diffusion, while $0 < \alpha < 1$ is sub-diffusion, representing faster and slower than normal diffusion, respectively; see Supplementary Sec. 6 for further details on the interpretation of the TAMSD and a physical justification of anomalous diffusion.

We find that GD dynamics exhibit anomalous diffusion and that these anomalous diffusive dynamics are non-stationary in a way that can be separated into two regimes. To emphasize this aspect, we divide TAMSD curves into two panels in Fig. 5: the upper one displays curves with smaller $t_w$ and the lower one with larger $t_w$. Both regimes contain sub-diffusion at small $\tau$ timescales and saturation ($\alpha \to 0$) at large $\tau$ timescales. The important difference is the first regime additionally features transient super-diffusion at intermediate $\tau$ timescales. This non-stationary anomalous diffusion is similar to that recently observed in realistic training scenarios[22]; a minor discrepancy is the apparent absence of saturation in these examples. However, some studies considering long timescales demonstrate the presence of saturation[24].

Taken together, these results indicate that optimizing on multifractal loss landscapes provides a unified mechanistic account of a wide range of deep learning dynamics. We now describe the conditions under which our analysis is applicable to real networks. Theoretically, the loss function must exhibit the multifractal geometric structure characteristic of our landscape model. Underlying our landscape model are the following assumptions: First, the pointwise Hölder exponent $H$ is continuously differentiable, ensuring that

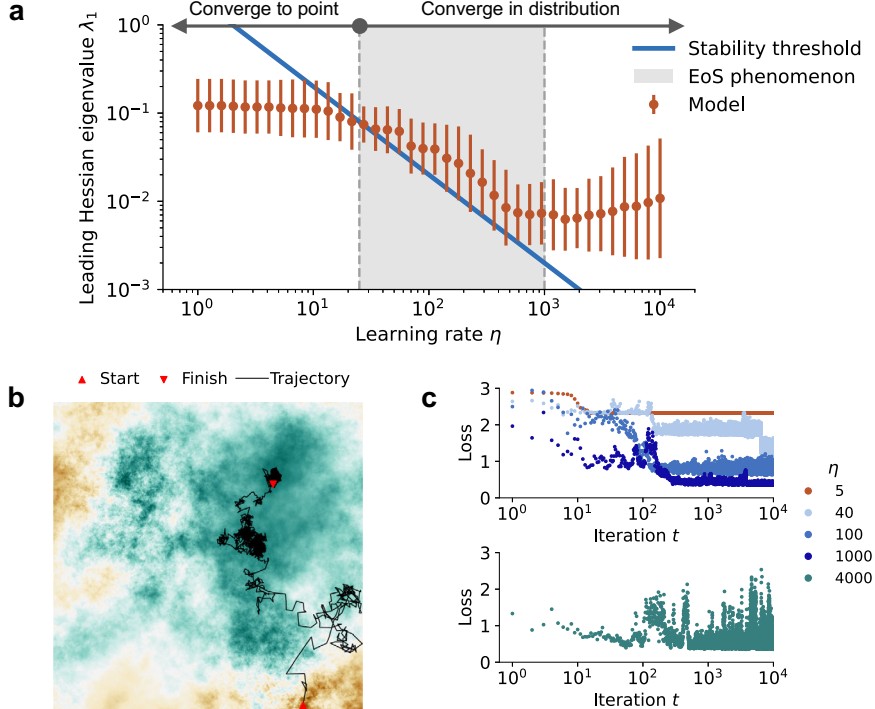

**Fig. 4 | Diverse dynamics of GD on a multifractal loss landscape. a** GD is simulated from 100 random initializations with 40 logarithmically spaced $\eta$ values. Dots and error bars represent the mean and 95% confidence intervals of the leading Hessian eigenvalue $\lambda_1$ in the last 100 iterations of each 10,000 iteration trajectory. The blue line delineates the stability threshold at $\lambda_1 = 2/\eta$. When GD is stable (i.e., $\lambda_1 < 2/\eta$), it converges to a local minimum of $L$. When GD is unstable (i.e., $\lambda_1 > 2/\eta$), it converges in distribution. Over more than a decade, dynamics exhibit the EoS phenomenon (shaded region): $\lambda$ is at or just above $2/\eta$, regardless of initialization.

These $\eta$ values represent practical values for realistic network training[29]. **b** Visualization of a GD trajectory with $\eta = 1000$. **c** Loss evolution across iterations obtained by tracking GD trajectories of varying learning rates $\eta$. The blue curves represent a wide range of learning rates that produce the EoS phenomenon; specifically, the dark blue curve corresponds to the trajectory in (**b**). The red and green curves represent learning rates outside the regime of the EoS phenomenon. While the red curve corresponds to a stable trajectory where loss monotonically decreases, the green curve corresponds to an unstable trajectory with large fluctuations in loss.

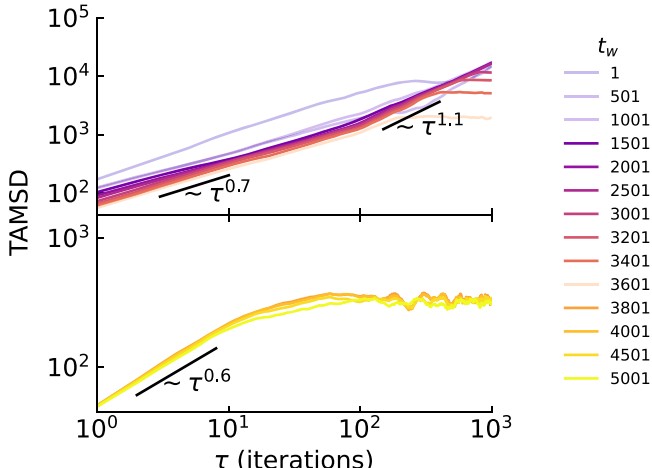

**Fig. 5 | Anomalous diffusive dynamics.** Time-averaged mean squared displacement (TAMSD) calculated for the black trajectory in Fig. 4b using a window of size $T = 3000$ and a systematic progression of waiting times $t_w$. To emphasize the non-stationary dynamics, we separate the initial transient super-diffusive regime (upper panel) from the subsequent sub-diffusive regime (lower panel). Eye-guides are provided to indicate approximate diffusion exponents $\alpha$. For visual clarity, we make some curves in the upper panel transparent. Note that these faded curves all still contain a super-diffusive segment with $\alpha > 1$.

roughness changes smoothly across space; see Supplementary Sec. 11 where, based on this requirement, we construct many different landscapes and demonstrate that it produces similar dynamics (Fig. S15). Second, the pointwise Hölder exponent $H$ should

exhibit a wide range of values. We find this is necessary to achieve a broad edge-of-chaos regime exhibiting the EoS phenomenon; the larger the range of $H$, the more extended and robust this regime is. In contrast, when the range is very small, as in monofractal and random landscapes and landscape models considered in previous machine learning literature, both the EoS phenomenon and the related edge of chaos are notably absent (Fig. S14). Because the true loss landscape is influenced by many factors, such as data properties and architectural characteristics, in a highly complex manner[65], it is mathematically challenging to determine the contributions of these factors to the landscape's geometrical structure. Consequently, previous investigations into these effects have relied on empirical approaches[38,39,55,65] showing, for instance, that the complexity of the loss landscape increases with task difficulty[65] but decreases with network width[55]. Similarly, our empirical findings corroborate the general applicability of our theoretical analysis when the loss landscape, specifically a two-dimensional section of it[55], exhibits multifractal characteristics, as indicated by a broad multifractal singularity spectrum, but not applicable otherwise; see Supplementary Sec. 4 for extensive real-network experiments supporting this.

## Fractional diffusion theory

We next adapt fractional formalisms used in the study of non-equilibrium physical systems[47] to formulate a fractional diffusion theory capable of reproducing and explaining the non-stationary anomalous diffusive dynamics as demonstrated above. Based on this theory, we then illustrate the computational advantages of such dynamics in the context of network optimization or learning. In particular, we apply the overdamped fractional Langevin equation[66,67],

focusing on the 1-D case for analytic tractability:

$$\Gamma(2\mathcal{H}-1)\,{}^{C}D_t^{2-2\mathcal{H}}\theta(t) = -\eta\frac{dV}{d\theta}(\theta(t)) + \sigma\xi_{\mathcal{H}}(t), \tag{10}$$

where $\mathcal{H} = \max\{1/2, H\}$, $H$ is the pointwise Hölder exponent of the loss landscape $L$, $\Gamma$ is the gamma function, ${}^{C}D_t^{\alpha}$ is the Caputo fractional derivative defined as[68]:

$$ {}^{C}D_t^{\alpha}\theta(t) \equiv \frac{1}{\Gamma(n-\alpha)}\int_0^t \frac{\theta^{(n)}(t')dt'}{(t-t')^{\alpha+1-n}} \tag{11}$$

where $n$ is the smallest integer strictly greater than $\alpha$ and $\theta^{(n)} \equiv d^n\theta/dt^n$, $V$ is a potential, $\xi_{\mathcal{H}} = \dot{B}_{\mathcal{H}}$ is the distributional derivative of the fractional Brownian process (i.e., fractional Gaussian noise) and $\sigma$ is the noise strength given by

$$\sigma = \begin{cases} \eta\sqrt{\frac{2}{\beta}}, & \text{for } H \in (0, 1/2] \\ \frac{\eta}{\sqrt{\beta H(2H-1)}}, & \text{for } H \in (1/2, 1), \end{cases} \tag{12}$$

where $\beta$ is an inverse temperature parameter.

We summarize the main ideas underlying equation (10); for a comprehensive theoretical analysis of large learning-rate GD in our multifractal model, see the derivation in Supplementary Sec. 8 and Fig. S11. The right-hand side of equation (10) can be understood as a formal decomposition of the loss landscape into large-scale, $V$, and small-scale, $\xi_H$, components. The representation of small-scale components as fractional Gaussian noise $\xi_H$ follows recent results on chaotic GD, suggesting that the gradient of small-scale components can be approximated by additive noise when the learning rate is sufficiently large[28]. Through conventional statistical mechanics analysis[69], the presence of fractional Gaussian noise implies a power-law memory term via the fluctuation-dissipation theorem. This term approximates the Caputo fractional derivative, resulting in the continuous-time fractional differential equation for iterative GD optimization. The utility of this approximation is that it enables the analytic derivation of various theoretical properties, as we demonstrate below. It is important to note that the fractional derivative, $\Gamma(2\mathcal{H}-1)\,{}^{C}D_t^{2-2\mathcal{H}}$, reduces to the regular derivative, $\frac{d}{dt}$, when $H \in (0, 1/2]$. As a result, equation (10) simplifies to the overdamped Langevin equation, which is a classical model of stochastic gradient descent (SGD)[18,19]. For $H \in (1/2, 1)$, equation (10) is a non-Markovian extension; such temporal correlations in the GD trajectory fundamentally arise from spatial correlations in $L$ (see the theoretical analysis in Supplementary Sec. 8, which further explicates this connection).

We now validate that equation (10) reproduces the nonstationary anomalous diffusion of GD. To capture the large-scale structure of $L$, we adopt minimal choices for $V$: First, we approximate the limiting basin as a harmonic potential via a second-order Taylor expansion, $\lambda\theta^2/2$ for constant $\lambda$. Second, we approximate the non-monotonic descent to the basin as a tilted washboard, $V_0\cos(2\pi\theta/\Theta) + F\theta$ for constant $V_0$, $\Theta$, and $F$. It is worth noting that this potential has previously been used to study super-diffusion in the fractional Langevin equation[70].

The harmonic component affects dynamics at large $t$ when the optimizer is in the limiting basin (i.e., the second regime of Fig. 5). Assuming a harmonic potential in equation (10) and large $T \gg \tau$, the TAMSD has an analytic expression[71]:

$$\overline{\delta^2}(t_w, \tau) \approx 2\langle\theta_\infty^2\rangle\left[1 - E_{2-2\mathcal{H}}\left(-\frac{\eta\lambda\tau^{2-2\mathcal{H}}}{\Gamma(2\mathcal{H}-1)}\right)\right], \tag{13}$$

where $\langle\theta_\infty^2\rangle = \eta/\beta\lambda$ is the variance of the equilibrium distribution. Here, $E_\kappa$ is the Mittag-Leffler function defined, for $\kappa > 0$, by the series

expansion $E_\kappa(z) = \sum_{n=0}^{\infty} z^n/\Gamma(1+\kappa n)$. Moreover, since $E_\kappa(z) = -\sum_{n=1}^{\infty} z^{-n}/\Gamma(1-\kappa n)$ for $z \to -\infty$:

$$\overline{\delta^2}(t_w, \tau) \sim \begin{cases} \tau^{2-2\mathcal{H}}, & \text{for } \tau \to 0 \\ \tau^0, & \text{for } \tau \to \infty. \end{cases} \tag{14}$$

Thus, equation (10) predicts a diffusion exponent $\alpha(H) = \min\{1, 2 - 2H\}$ for small $\tau$ timescales. This implies that GD dynamics exhibit normal diffusion in rougher basins with $H \in (0, 1/2]$ and sub-diffusion in smoother basins with $H \in (1/2, 1)$. At large $\tau$ timescales, equation (10) predicts saturation. To validate these predictions, we fit equation (13) to experimental TAMSD curves and estimate the diffusion exponent in the first decade of $\tau$ (see "Methods" for details on the fitting procedure). As shown in Fig. 6a–b, there is good agreement between our model and the fractional diffusion theory.

On the other hand, the tilted washboard component affects dynamics at small $t$ while the optimizer approaches the limiting basin (i.e., the first regime of Fig. 5). Since no analytical solution is available for a general potential[70], we numerically solve equation (10) and then calculate the TAMSD (see "Methods" for details on the numerical scheme). Under choices of simulation parameters that approximate the black trajectory in Fig. 4b (e.g., 10,000 iterations, learning rate is 1000, starting point is around 700 units away from the limiting basin, etc.; see "Methods" for all parameters), we obtain TAMSD curves (Fig. 6c) that are remarkably similar to that of our model (Fig. 5). Notably, the curves include a transition from an initial transient super-diffusive regime to an ensuing sub-diffusive regime.

In summary, the consistency of the fractional diffusion theory with our multifractal model in terms of the TAMSD (as well as temporal correlation behavior of the optimization process, see Supplementary Sec. 9) indicates that it accurately describes the learning dynamics of GD on the loss landscape. In this sense, it is an operational theory of GD training. It is important to note that, unlike previous Langevin equation-based theories of deep learning dynamics that only focus on the local solution space at the end of training, our fractional diffusion theory is able to explain the time-varying nature of anomalous diffusion throughout the entire training process. For further physical justification of the anomalous diffusion of GD in a multifractal loss landscape, as well as specific features of the TAMSD, see Supplementary Sec. 6.

**Fractional diffusion facilitates optimization**

Based on our fractional diffusion theory, we now illustrate that these non-stationary anomalous diffusive dynamics are advantageous for deep learning.

Indeed, the initial super-diffusion enables fast exploration of the loss landscape, expediting the search for a low-loss basin. The ensuing sub-diffusion then facilitates exploitation. Crucially, this sub-diffusion is slower in smoother basins, as indicated by equation (14) derived from our fractional diffusion theory. Simultaneously, equation (5), derived from the geometrical properties of the multifractal loss landscape, reveals that smoother basins are populated by clusters of well-connected flatter minima. Combining these two theoretical aspects, it is interesting to note that the learning dynamics exhibit an adaptive nature akin to a landscape-dependent annealing process wherein the optimizer automatically tends towards well-connected flatter minima by diffusing more slowly in their vicinity. This adaptive behavior is particularly pronounced in the smooth limit $H \to 1^-$. Here, our fractional diffusion theory predicts that the time-averaged displacement of network parameters decreases polynomially with the flatness of a solution ($\overline{\delta^2} \sim F^{-4}$, see Supplementary Sec. 7 for the mathematical derivation). These results indicate that the development of an adaptive learning algorithm that leverages these landscape-aware properties may be promising. We should also note that a similar relation between the time-averaged variance of network parameters and a non-locally

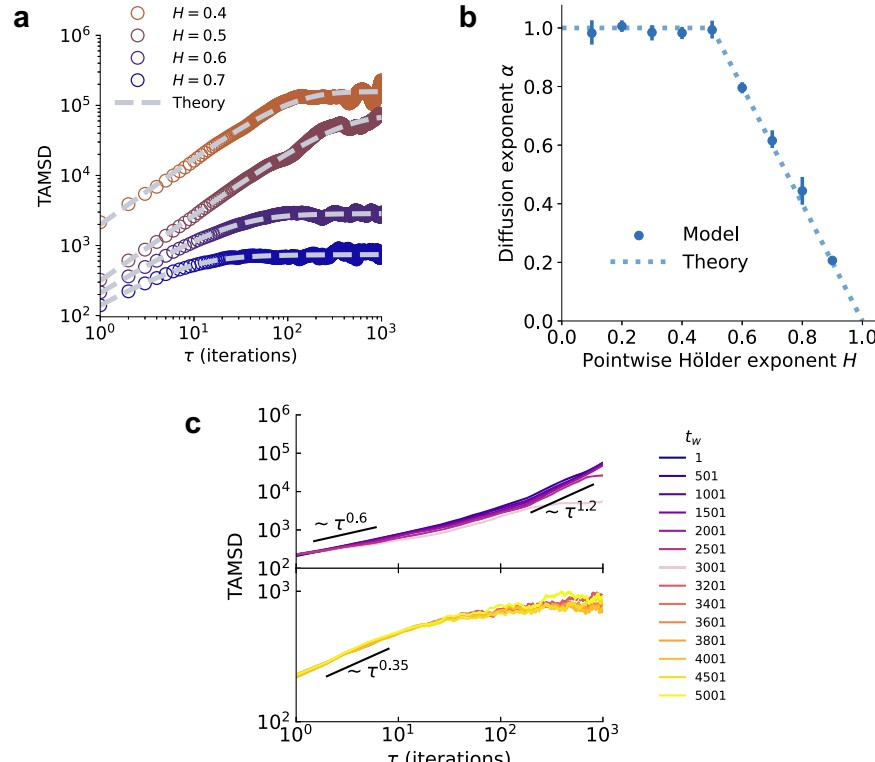

**Fig. 6 | Experimental validation of the fractional diffusion theory. a** Theoretical predictions of the time-averaged mean squared displacement (TAMSD) in equation (13) (gray dashed lines) are fit to TAMSD curves ($T = 1000$) for GD trajectories on the loss landscape $L$ constructed from $B_H$ with constant $H \in \{0.4, 0.5, 0.6, 0.7\}$. TAMSD curves in the second regime, where equation (13) is applicable, are obtained for analysis. For visual clarity, we separate the curves by multiplying by arbitrary factors; note that this does not change the diffusion exponent at small lag times $\tau$. **b** For each $L$ constructed from $B_H$ with constant $H \in \{0.1, 0.2, ..., 0.9\}$, we simulate 5 random initializations of GD. The diffusion exponent $\alpha$ is estimated from least-squares regression on the power law segment in the first decade of $\tau$ for TAMSD curves in the second regime. Dots and error bars represent mean and 95% confidence interval calculated from 10,000 bootstrap resamples, respectively. The dotted line corresponds to the theoretical prediction in equation (14). **c** TAMSD calculated from the numerical solution of equation (10) using a window of size $T = 3000$ and a systematic progression of waiting times $t_w$. Eye-guides indicate approximate diffusion exponents $\alpha$.

defined flatness was empirically found during SGD training[34]. Subsequently, this relation was related to a preference for flatness via properties of SGD noise. Nevertheless, our analysis differs in two key aspects: First, we forgo assumptions about the structure of SGD noise as GD is deterministic. Second, our overall analysis does not focus solely on the final solution space but instead consider dynamics throughout the entire training procedure.

To further characterize learning dynamics, we examine the convergence behavior of equation (10) using Theorem 2 of ref. 72, which applies at large $t$ when $V$ (i.e., basin of the loss landscape) is locally harmonic; overall, it indicates that the optimizer is more likely to converge towards higher-volume solution spaces. The theorem states that the solution to equation (10) converges to a stationary Gaussian process whose power spectral density, which characterizes its temporal correlations, has a particular analytic expression; as shown in Fig. S13, this power spectral density aligns with our multifractal model. Furthermore, its probability distribution converges to the Gibbs distribution:

$$p(\boldsymbol{\theta}) \propto \exp\left(-\frac{\beta V(\boldsymbol{\theta})}{\eta}\right). \qquad (15)$$

This has been conjectured to still be valid for higher dimensions and arbitrary potentials. This Gibbs equilibrium distribution implies that the probability is concentrated in basins rather than individual minima, given that the learning rate in the edge-chaotic regime where the EoS phenomenon occurs, is too high to resolve the small-scale details of the loss landscape. Moreover, for large $\beta/\eta$ we can apply the Laplace

approximation to determine the ratio of probability masses between two basins centered at $\boldsymbol{\theta}_A$ and $\boldsymbol{\theta}_B$:

$$\frac{P(\boldsymbol{\theta} \in \text{basin } A)}{P(\boldsymbol{\theta} \in \text{basin } B)} \approx \sqrt{\frac{\det \mathbf{H}(\boldsymbol{\theta}_B)}{\det \mathbf{H}(\boldsymbol{\theta}_A)}} \exp\left[\frac{\beta}{\eta}\left(V(\boldsymbol{\theta}_A) - V(\boldsymbol{\theta}_B)\right)\right]. \qquad (16)$$

In cases where two basins are nearly degenerate (i.e., $V(\boldsymbol{\theta}_A) \approx V(\boldsymbol{\theta}_B)$), the first term becomes more important, indicating a preference for the flatter basin where $\det \mathbf{H}$ is smaller. This preference translates into a prediction for higher-volume solution spaces in the sense that the connected set, whose potential is similar to the minimum, is larger; such a solution space inherently possesses a greater capacity for hosting a larger cluster of highly degenerate solutions. The result proves functionally advantageous for both the optimizability of a network (facilitating the finding of good solutions) and the navigability of the solution space (enabling the exploration of diverse solutions). We note that the Gibbs equilibrium distribution has previously been derived in an analysis of SGD using a conventional Ornstein-Uhlenbeck model[18,19]; however, in this context, the energy term of the Gibbs distribution changes from the potential $V$ to the overall loss $L$, causing peaks in probability within individual minima, rather than basins. However, peaks in probability within basins have been empirically validated in the equilibrium distribution of chaotic GD in complex loss landscapes[28], aligning with our analysis.

We proceed to analyze metastability through the lens of the mean first passage time $\langle \tau \rangle$, defined as the mean time taken for the optimizer to escape a basin; this analysis serves to further demonstrate preference for smoother basins containing flatter minima. For $H \in (0, 1/2]$,

assuming the set-up of the 1-D Kramer's escape problem[73], the mean first passage time is given by:

$$\langle \tau \rangle = \frac{2\pi}{\sqrt{\mathbf{H}(\theta_A)|\mathbf{H}(\theta_B)|}} \exp\left[\frac{\beta}{\eta}(V(\theta_B) - V(\theta_A))\right], \quad (17)$$

where $\mathbf{H}$ is the Hessian (which is simply a constant in 1-D), $\theta_A$ is the basin minimum, and $\theta_B$ is the local maximum via which the optimizer escapes. A generalization to higher dimensions was derived in a work modeling SGD with the overdamped Langevin equation[74]. Notably, $\langle \tau \rangle$ is finite for $H \in (0, 1/2]$ across any dimensions. However, for the non-Markovian case of $H \in (1/2, 1)$, a different set-up has been used to maintain analytic tractability[75]: $V(\theta) = \lambda\theta^2/2$ for $\theta < \Theta$, where $\Theta$ is constant, and $V(\theta) = -\infty$ otherwise. Assuming $\Theta \gg \sqrt{1/\beta\lambda}$, the probability that the optimizer remains in the harmonic cusp after a time $t$ can be approximated:

$$P(t) \approx \left[E_{2-2H}\left(-\frac{\eta\lambda t^{2-2H}}{\Gamma(2\mathcal{H}-1)}\right)\right]^{s_0}, \quad (18)$$

where $s_0$ is a constant that depends on $\eta$, $\beta$, and the barrier height $\lambda\theta^2/2$. The mean time for the optimizer to fall out of the harmonic cusp is thus divergent:

$$\langle \tau \rangle = \int_0^\infty P(t)dt = \infty. \quad (19)$$

For deeper insights, consider the large $t$ limit where equation (18) behaves as $P(t) \sim 1/t^{s_0\alpha}$, where $\alpha = 2 - 2H$ is the diffusion exponent. The power-law exponent reveals the important role of anomalous diffusion in the exploitation ability of GD, since the slower diffusion rate (smaller $\alpha$) in smoother basins results in longer escape times (see Fig. S16 in Supplementary Sec. 12 for further numerical evidence of this). Merging the metastability results of the two cases, we overall conclude that the optimizer tends to avoid rougher basins (i.e., $H \in (0, 1/2]$) as $\langle \tau \rangle$ is finite, and favors smoother basins (i.e., $H \in (1/2, 1)$) as $\langle \tau \rangle$ is infinite.

In summary, through developing the fractional diffusion theory, we elucidate that the non-stationary anomalous diffusive dynamics are functionally advantageous for deep learning. According to the theory, these dynamics harness landscape-dependent annealing-like properties, ultimately steering the optimizer towards large and smooth solution spaces which house well-connected flatter minima, as delineated by equation (5). Given that generalization has been strongly correlated with both local flatness[24,32–35] and global connectivity[14], our model effectively explains how the learning dynamics of GD can attain generalizing solutions in deep learning.

## Discussion

In this study, we have presented a theoretical framework that elucidates the mechanistic links between realistic geometrical signatures (including multiscale structure, highly degenerate minima separated by low barriers, and their clustering) of loss landscapes with actual optimization dynamics in deep learning (the EoS phenomenon, edge of chaos, and non-stationary anomalous diffusion). In addition, we have revealed that the GD optimizer exploits the complexities of loss landscapes in a landscape-dependent annealing-like manner. This enables the optimizer to navigate towards larger and smoother solution spaces that house well-connected flatter minima, thus facilitating generalization. Our framework thus provides a unifying, non-equilibrium statistical mechanics approach for understanding deep learning.

Beyond reconciling the aforementioned established properties of real loss landscapes, our model also predicts several properties that emerge through its multifractal structure. For instance, the distribution of curvatures across minima can be broad due to significant heterogeneity in the multifractal landscape. Furthermore, while our model exhibits clustering of these minima as consistent with previous theoretical analyses[36,37], we specifically predict a fractal organization where the number of minima in a cluster grows as a power-law of its size. Additionally, our model predicts that the roughness of the loss landscape at the cluster correlates with the expected curvature of minima therein, such that clusters in smoother regions house well-connected flatter minima. This predicted correlation between smoothness and flatness may explain why global connectivity and local curvature are both associated with generalization[14]. Empirical investigations of these emergent properties may be conducted using techniques developed in the study of energy landscapes, such as basin-hopping methods to locate local minima[38,39]. Although we have demonstrated that two-dimensional sections of true loss landscapes exhibit multifractal structure, further validation of multifractality in higher dimensions would require the development of computationally efficient multifractal formalisms (see e.g.,[76] and references therein). Alternatively, a theoretical analysis may follow an appropriate spin-glass characterization of the loss landscape where, notably, a fractal phase is present[77]. Preliminary work also suggests the potential to extend state-of-the-art tools from topological data analysis (e.g., persistent homology) to evaluate the fractal geometry of the loss landscape[65]. Overall, our results suggest that the characteristics of the loss landscape, particularly in terms of the clustering of minima, their multiscale behavior, and curvatures, have important implications for understanding the navigability of the loss landscape, the breadth of the EoS phenomenon, and the generalization capacity of solutions.

Surprisingly, and even counter-intuitive, dynamical phenomena emerging in deep neural network optimization have garnered recent attention, including chaos[28,78] and the EoS phenomenon[29,40,53,79]. These phenomena alike have been investigated through the lens of loss landscape models that forgo unrealistic quadratic Taylor approximations, including two-scale and multiscale loss landscapes[28,40]. In contrast, our multifractal loss landscape encompasses both chaotic dynamics and the EoS phenomenon, where the leading Hessian eigenvalue hovers just above the stability threshold of 2/(learning rate) regardless of initialization and across a wide range of learning rates. While chaotic GD in a two-scale loss landscape exhibits a discrete transition in convergence behavior, from the equilibrium distribution peaking in small-scale minima to large-scale basins as learning rate increases[28], this transition is continuous in our model as it is multiscale in a continuum sense. Moreover, since we have demonstrated that the stability threshold coincides with the onset of chaos through analysis of the local Lyapunov exponent, our framework explicitly relates the EoS phenomenon to an edge of chaos that is extended across hyperparameter (e.g., learning rate) space; this shared extended regime broadens with the range of Hölder exponents in the multifractal loss landscape, but is notably absent in surrogate monofractal and random landscapes (see Supplementary Sec. 10 and 11 for these results).

Our model unveils a crucial insight: near the edge of chaos, the dynamics of network optimization exhibit a landscape dependence that yields significant computational advantages. Rather than converging indiscriminately to any local minimum with a small learning rate or diverging from the solution space with a large learning rate, the optimizer demonstrates roughness-dependent anomalous diffusion. This behavior enables convergence in distribution towards a solution space populated by well-connected flatter minima. Importantly, as the edge of chaos is extended—with its range increasing alongside the pointwise Hölder exponents that characterize multifractality—this enhanced generalization property does not require precise fine-tuning. It is worth noting that the concept of the edge of chaos has been proposed in previous theories of learning systems as an optimal setting for computation[60,62]. In deep learning, the edge of chaos has been applied to layerwise signal propagation in deep networks. However, in contrast to our results, the order-to-chaos boundary in that context is non-extended, requiring stringent fine-tuning to achieve computational benefits such as preventing vanishing and exploding gradients[63,64].

There is a large body of related work applying statistical mechanics approaches to understand optimization processes[6–17]. An important idea originating from the statistical mechanics formulation of learning is that the dynamical behavior of learning algorithms could be studied to understand the energy landscape itself[7,16]. In particular, Langevin dynamics as a theoretical framework has extended from early studies within this context[6] to more recent investigations of SGD[18,19]. Since empirical evidence of more complex stochastic behavior has emerged, such as anomalous diffusion[22,23] and heavy-tailed statistics[15], optimization dynamics with non-Gaussian heavy-tailed noise[21] and multiplicative noise[27] have been considered. The advantages of these dynamics for generalization are understood through heavy-tailed random matrix theory, which is deep-seated in statistical physics[15]. However, it is unclear whether such theories can explain the non-stationary nature of anomalous diffusion observed in deep learning (for example, see Fig. S17 in Supplementary Sec. 12 for an empirical comparison of optimization dynamics). In contrast, we have developed a fractional Langevin equation-based theory, in which stochasticity effectively arises from deterministic interactions with the multifractal structure of the loss landscape. Our fractional diffusion theory mechanistically relates non-stationary anomalous diffusive dynamics during training to the spatially-varying structure of the loss landscape. Specifically, the diffusion exponent depends on the Hölder exponent.

Furthermore, our framework elucidates the crucial functional advantage of such landscape-dependent dynamics in achieving well-connected, flatter solutions corresponding to better generalization[14]. This is exemplified by the prediction of our model that, in the smooth limit, the time-averaged displacement of network parameters decreases polynomially with the flatness of a solution. Future work could further leverage these landscape-aware annealing-like properties to develop improved optimization algorithms (see Fig. S18 and Supplementary Sec. 13, where we construct a simple algorithm using landscape-dependent anomalous diffusion as a proof of concept). It is worth noting that, as our study focused on GD, it is applicable to explaining the impressive generalization performances that have been found in large-batch and full-batch training scenarios, which are in some cases even comparable to small-batch SGD[80,81]. The insights gained may also be valuable for future investigations into whether SGD noise is dependent on the roughness of a multifractal loss landscape, providing a more complete picture of neural network training dynamics.

The emergence of complex dynamics from intricate geometrical structures is a common phenomenon in various complex systems. For instance, this is observed in the anomalous diffusion of macromolecules in fractal-structured chromosomes[82] and in the conformational states of proteins navigating rough energy landscapes[83]. We posit that our model and theory of network optimization might have general applicability to understanding the functional advantages of such emergent dynamics in certain complex systems. For example, statistical analyses of biological fitness landscapes have similarly connected spatial correlations in the landscape to fractality[48]. Our analysis could provide a framework for understanding how biological fitness landscapes, despite their high dimension and extreme ruggedness, remain easily navigable[49]. Consistent with our findings, the complexity of such landscapes may not impede navigation or optimization processes, but may instead facilitate them.

## Methods

### Simulating GD on a multifractal loss landscape

We generate multifractal loss landscapes using the FracLab toolbox in MATLAB[84], which creates a realization of the random Gaussian function $B_H$ with a user-defined continuously differentiable pointwise Hölder exponent $H$ and covariance given by equation (4). Note that $B_H$ is specifically the multifractional Brownian surface[51,52] (see Supplementary Sec. 1 for further details). We discretize $B_H$ on an $N \times N$ Cartesian grid in the domain $[0, 1] \times [0, 1]$. For simulations of GD, we re-scale the Cartesian grid to have unit spacing. We then calculate the loss gradient $\nabla L = (L_x, L_y)$ at points on the Cartesian grid by central differences, except for at the boundaries where forward or backward differences are necessary. To determine the gradient at an arbitrary point, we apply bilinear interpolation. As a result, the partial derivatives of $L$ are continuous, implying the differentiability of $L$. During simulations of GD learning dynamics, we impose symmetric boundary conditions such that an optimizer is reflected at the boundary.

In the representative example of the main text, we use pointwise Hölder exponent $H(x, y) = 0.5 + 0.2 \sin(\pi x / 512) \cos(3\pi y / 2048)$ on a $1024 \times 1024$ Cartesian grid because it contains a relatively broad range of values for illustration purposes (see Supplementary Sec. 11 for additional realizations, including other choices of $H$, to demonstrate our results are general). To characterize the multifractal nature of $L$, we apply the wavelet leader multifractal formalism which is implemented in the PLBMF toolbox on MATLAB[50].

### Derivation of the expected total curvature

We first illustrate that $\| \mathbf{H} \|_F^2$ is an appropriate scalar index for total curvature. As the Hessian matrix $\mathbf{H}$ is real and symmetric, it is normal. Hence, the (squared) Frobenius norm of the Hessian matrix can be expressed in terms of its eigenvalues as $\| \mathbf{H} \|_F^2 = \sum_{i=1}^n \lambda_i^2$, where $\lambda_i$ are the Hessian eigenvalues satisfying $|\lambda_1| \geq |\lambda_2| \geq \cdots \geq |\lambda_n|$. Since the Hessian eigenvalues measure the curvature along the principal directions corresponding to the eigenvectors of $\mathbf{H}$, then $\| \mathbf{H} \|_F^2$ serves as a scalar index for total curvature.

We now derive equation (5), which relates the expected curvature of the multifractal loss landscape $L$ to its pointwise Hölder exponent. Consider a multifractal loss landscape $L$ constructed from the 2-D multifractional Brownian surface $B_H$. According to the aforementioned construction, $B_H$ is discretized on a fine $N \times N$ Cartesian grid with a spacing denoted as $\chi = 1/N$. Applying the centered difference approximation for the second derivative of $L$ gives:

$$L_{xx}(x,y) \approx \frac{B_H(x+\chi, y) - B_H(x,y) - B_H(x,y) + B_H(x-\chi, y)}{\chi^2}, \quad (20)$$

$$L_{yy}(x,y) \approx \frac{B_H(x, y+\chi) - B_H(x,y) - B_H(x,y) + B_H(x, y-\chi)}{\chi^2}, \quad (21)$$

$$\begin{aligned} L_{xy}(x,y) &\approx \frac{B_H(x+\chi, y+\chi) - B_H(x+\chi, y-\chi) - B_H(x-\chi, y+\chi) + B_H(x-\chi, y-\chi)}{4\chi^2} \\ &= L_{yx}(x,y). \end{aligned}$$
$$(22)$$

Assuming that the pointwise Hölder exponent $H$ is slowly varying such that $\frac{\partial H}{\partial \theta_i} \chi \ll H$, then the pointwise Hölder exponents at adjacent grid points are approximately equal. Thus, we can evaluate the expected value of the squares of the second derivatives of $L$ using the covariance of the multifractional Brownian motion $B_H$ (equation (4)). Summing these expectations yields the expected total curvature:

$$\langle \| \mathbf{H} \|_F^2 \rangle = \langle L_{xx}^2 \rangle + \langle L_{xy}^2 \rangle + \langle L_{yx}^2 \rangle + \langle L_{yy}^2 \rangle \propto r(H) \chi^{2H-2}, \quad (23)$$

where $r(H) \equiv 16 - 2^{2H+2} + 2^{2H} - 2^{3H-1}$. Generalization to higher dimensions is straightforward. Note that the constant of proportionality in equation (23) is $D_n(H, H)$, where $D_n : \mathbb{R} \times \mathbb{R} \to \mathbb{R}$ is a deterministic function that depends on the dimensionality $n$[52].

### Clustering analysis

To quantify the clustering of low-loss solutions in the multifractal loss landscape $L$, we choose a reference solution (visualized as a dot in Fig. 2c) and count the number of other low-loss solutions $N$ within a Euclidean distance $R$ for an array of logarithmically-spaced distance values. We repeat and average this calculation across all low-loss

solutions within a basin, and additionally repeat for multiple basins (as shown in Fig. 2d). As a result, we find that $N$ grows as a power-law function of $R$. To measure the power-law exponent $d_f$, we use least-squares regression on a large segment that is locally straight on logarithmic axes. Moreover, we estimate the size $R_0$ of the cluster as the distance $R$ where the residuals of the fitted power-law begin to diverge.

## Characterizing the loss landscape of deep neural networks

We train a VGG-16 network on the CIFAR-10 dataset for 50 epochs using cross-entropy loss and SGD with a batch size of 64 and a learning rate of 0.001. After a number of epochs, we evaluate the training loss on a fixed mini-batch across a two-dimensional plane spanned by filter-normalized random directions[55]. Specifically, we sample the loss on a $257 \times 257$ grid with side lengths equal to 2.4 times the filter norm. This loss surface directly relates to the optimization dynamics, as its gradient governs the update in the next iteration. We use the wavelet leader multifractal formalism in the PLBMF toolbox to calculate multifractal singularity spectra. In Supplementary Sec. 4, to investigate the robustness of the results, we perform the same characterizations using the ResNet architecture, the FashionMNIST dataset, and the Adam optimizer, totaling 8 different training configurations. Although we do not tune hyperparameters, all configurations achieve reasonable test accuracy, as demonstrated by learning curves in Fig. S4.

## Determination of chaos

As a test of chaos, we measure the maximal Lyapunov exponent. For two trajectories in phase space initially separated by a vector $\delta\boldsymbol{\theta}_t$ at time $t$, it is defined as:

$$\lambda \equiv \lim_{t \to \infty} \lim_{\|\delta\boldsymbol{\theta}_0\| \to 0} \frac{1}{t} \ln \frac{\|\delta\boldsymbol{\theta}_t\|}{\|\delta\boldsymbol{\theta}_0\|}. \tag{24}$$

We use Sprott's algorithm[59] to estimate the MLE, which is based on equation (8). We implement it as follows: For a fixed $\eta$, we simulate GD from a random initialization. Every 10 iterations, we generate a branch of the GD trajectory starting at a small perturbation $\delta\boldsymbol{\theta}_0$ from the original (e.g., 0.1 grid units). We measure $\delta\boldsymbol{\theta}_t$ as the separation between the original and the branch after a short time (e.g., 10 iterations). We find that the cumulative average of $\frac{1}{t} \ln \frac{\|\delta\boldsymbol{\theta}_t\|}{\|\delta\boldsymbol{\theta}_0\|}$ reliably converges after 1000 iterations (i.e., 100 branches). This is taken as one estimate of the MLE. We repeat this process for 100 random initializations and perform 10,000 bootstrap resamples to calculate the mean and 95% confidence interval of the MLE.

## Numerical solution of the overdamped fractional Langevin equation

The integral formulation of the overdamped fractional Langevin equation (equation (10)) is written as[85]:

$$\Gamma(\alpha)\Gamma(1-\alpha)[\theta(t) - \theta(0)] = -\eta \int_0^t (t-t')^{\alpha-1} \frac{dV}{d\theta}(\theta(t')) \, dt' + \sigma \int_0^t (t-t')^{\alpha-1} \, dB_H, \tag{25}$$

for $\alpha = 2 - 2H \in (0, 1)$ (i.e., $H \in (1/2, 1)$). Note that for $H \in (0, 1/2]$ in our model, equation (10) becomes the memoryless Langevin equation which can be integrated in accordance to the well-known Euler-Maruyama method. Thus, we focus on the non-Markovian case here. To numerically integrate equation (25), we apply a recently proposed scheme which converges in strong sense[85]:

$$\theta_t = \theta_0 - \frac{\eta}{\Gamma(2H-1)\Gamma(3-2H)} \sum_{j=1}^{t} \frac{dV}{d\theta}(\theta_{j-1})[(t-j+1)^{\alpha} - (t-j)^{\alpha}] + \gamma B_{1-H}, \tag{26}$$

where $\theta_t \equiv \theta(t)$, $\gamma = \eta\sqrt{2/\beta\Gamma(3-2H)\Gamma(2H-1)}$. It is important to note that the sum encodes the history of all previous iterations; the gradient

terms of the summation can be stored in memory. We simulate the fractional Brownian process $B_{1-H}$ using the Davies-Harte method[86]. Figure S12 demonstrates the utility of this scheme for validating theoretical predictions of the overdamped fractional Langevin equation.

## Fitting results of the fractional diffusion theory to experimental TAMSD curves

For experimental TAMSD curves in the second regime, we fit the analytic approximation (equation (13)) derived from the fractional diffusion theory. Specifically, we determine values for the variance of the equilibrium distribution $\langle\theta_\infty^2\rangle$, the curvature $\lambda$, and the Hölder exponent $\mathcal{H}$ by least squares regression. These parameters determine the plateau value, the characteristic plateau time, and the rate at which the TAMSD grows, respectively. For a further example of this procedure, see Supplementary Sec. 8.

For experimental TAMSD curves in the first regime, the fractional diffusion theory has no analytic result. Thus, we implement the numerical solution in equation (26) and then calculate the TAMSD. Assuming the potential $V$ representing the large-scale structure of the loss landscape is a symmetric piecewise combination of tilted washboard components funneling into a centered harmonic component:

$$V(\theta) = \begin{cases} -V_0 \cos(2\pi\theta/\Theta) - F\theta, & \text{for } \theta < w \\ \lambda\theta^2/2, & \text{for } -w < \theta < w \\ V_0 \cos(2\pi\theta/\Theta) + F\theta, & \text{for } \theta > w, \end{cases} \tag{27}$$

we must specify the following landscape-based parameters: the width $w$ and sharpness $\lambda$ of the harmonic component; the amplitude $V_0$, period $\Theta$, and bias $F$ of the tilted washboard component. We must also specify the following optimizer-based parameters: the initialization $\theta_0$, number of iterations $T$, learning rate $\eta$, the pointwise Hölder exponent $H$, and inverse temperature $\beta$. Note that decreasing $\beta$ increases the noise strength (and thus its destabilizing effect) relative to the gradient of the potential (which, for a harmonic potential, has a stabilizing or confining effect). In the particular example of Fig. 6c, we choose these parameters to roughly reflect the scenario shown as the black trajectory in Fig. 4b: $w = 2 \times 10^2$, $\lambda = 5 \times 10^{-4}$, $F = 10^{-2}$, $\Theta = 10^2$, $V_0 = 10^{-1}$, $\theta_0 = -7 \times 10^2$, $T = 10^4$, $\eta = 10^3$, $H = 0.7$, $\beta = 5 \times 10^3$. Note that the goal of the example was not to fit to Fig. 5, rather it was to demonstrate that the fractional diffusion theory can produce the non-stationary anomalous diffusion of our model. Fitting can be accomplished by applying Bayesian methods or genetic algorithms.

## Data availability

All data generated in this study have been deposited in the Zenodo database (https://doi.org/10.5281/zenodo.14997499)[87]. The neural network training data used in this study was downloaded from https://www.cs.toronto.edu/~kriz/cifar.html. Source data are provided with this paper.

## Code availability

The code for simulations and analyses of gradient descent on a multifractal loss landscape is available without restrictions on Github (https://doi.org/10.5281/zenodo.14997616)[88].

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

## Acknowledgements

This work was supported by the Australian Research Council (grant no DP160104368).

## Author contributions

A.L. and P.G. designed the study, performed the research, and wrote the paper.

## Competing interests

The authors declare no competing interests.
