## [Transparent Peer Review file · Nature Communications]

Optimization on multifractal loss landscapes explains a diverse range of geometrical and dynamical properties of deep learning

Corresponding Author: Professor Pulin Gong

Version 0:

Reviewer comments:

Reviewer #1

(Remarks to the Author)

This is a review of "Optimization on multifractal loss landscapes explains diverse range of geometrical and dynamical properties of deep learning."

The paper considers some of the recent work in machine learning having to do with learning with neural network models, and it presents a multifractal model that reproduces many of the properties. There has been a lot of work within machine learning on this general topic. Overall, the paper presents a number of interesting results that will probably be good for the machine learning community to know about. They will probably not be surprising to the physics community. On the other hand, the paper presents the results by comparing to relatively straw men baselines (common in the machine literature, but naive from the statistical physics perspective). So, the paper actually reads more like a review of statistical physics type approaches, without any acknowledgement of many of the results in that statistical physics literature, instead only comparing to recent machine learning discussion of the topic.

With regard to the work in machine learning on this topic (which is the general framing of the paper), there is a lot of insight, there is also a large theory-practice disconnect. For example, most theory about SGD says very little about optimization dynamics, either since it is formulated for convex problems (until rather recently) or since it is on very idealized e.g. infinitely wide models (more recently). Some of the references below and other papers by those authors consider perspectives on optimization that are stronger baselines.

On the other hand, with regard to work in the statistical mechanics of learning, there is a large body of work that is ignored, and I should note that the work that is mentioned (e.g., the Bahri review, Ref 33) is not all of the statistical mechanics of learning, but only the part that is less relevant for the problems considered in that paper. Much more relevant is older work such as:

Seung et al., "Statistical mechanics of learning from examples," Physical Review A, 1992.
Haussler et al., "Rigorous learning curve bounds from statistical mechanics," Machine Learning, 1996.
Watkin et al., "The statistical mechanics of learning a rule," Rev. Mod. Phys., 1993.
Engel et al. "Statistical mechanics of learning," Cambridge University Press, 2001.

but there is also more recent work such as:

"Generalization Bounds using Lower Tail Exponents in Stochastic Optimizers," Hodgkinson et al., ICML22
"Taxonomizing local versus global structure in neural network loss landscapes," Yang et al., NeurIPS21
"Multiplicative noise and heavy tails in stochastic optimization," L. Hodgkinson et al., ICML21
"Implicit Self-Regularization in Deep Neural Networks: Evidence from Random Matrix Theory and Implications for Learning," C. H. Martin et al., JMLR21

Note that most of that work does not consider fractal/multifractal issues per se, so it doesn't undermine this paper. But that work does present results, either statistical mechanics formulations of the learning process or heavy tailed random matrix

approaches or loss landscape or optimization dynamics approaches that are much less straw men than much of the machine learning literature that is cited. There is also a body of work in multiple papers but summarized in the book:

Sornette. "Critical phenomena in natural sciences: chaos, fractals, selforganization and disorder: concepts and tools," Springer-Verlag, 2006.

some of which does consider fractality/multifractality.

I mention that since there are many comments throughout, e.g., starting with the abstract which says "... complexities of loss landscapes do not hinder optimization as conventionally thought," where the "conventionally" only refers to very recent work in machine learning that is ignorant about the older physics work. Similarly, edge of stability was well-known in the 90s statistical mechanics of learning, but ignored by the recent machine learning community. That's a statement about what the machine learning community chooses to pay attention to. But a paper like this should remedy that, especially since one of it's selling points is making connections with, e.g., non-equilibrium systems and other such complex physical systems and physical concepts, and being published in a non machine learning venue.

There are multiple technical issues that would be good to clarify, even if it is in the supplementary material. One example is why work with the Frobenius norm of the Hessian, as opposed to top-k eigenvalue sum or trace. The explanation is just that it is a metric, but how does it compare to related metrics. This is relevant not just as a technical quibble, but also since there is a lot of work on Hessian properties "explaining" good generalization, which is at most a partial explanation since it is a very local metric.

The figures were reasonably convincing that the suggested model explained many of the qualitative properties it aims to explain. It would be good, however, even if in an appendix to have more details about the models considered, variants to the model trained to different types of data, etc., i.e., "controls" (in science) or "ablations" in machine learning.

At line 180, it is said (correctly) that SGD analysis usually makes assumptions about l-smoothness, etc. I said above the this reads more like a review than a new paper since, e.g., that analysis is about convex problems, etc., which clearly are not applicable to the learning process. (Admittedly, machine learning papers before a few years ago did make such assumptions, but that doesn't mean that a paper like this should use that as a baseline. The Hodgkinson paper above considers a much more realistic baseline. To my mind, this contrasts with much of the edge of chaos type discussion which (even though not well placed in the physics literature, which knew about it long ago) does make what to my knowledge are new connections with fractality, in the machine learning community at least.

Line 256: I'm not sure about the explanation for curse of dimensionality. Also, the mention of "realistic examples" leads me to want more details on the details of the models presented in the figures.

I think that the value of the paper is putting a lot of work in a common setup. But I think that its effectiveness is hindered by not placing it well in the context of state of the art methods in the machine learning literature on optimization dynamics, loss landscapes, random matrix theory, etc., as well as state of the art methods from 90s style statistical physics of learning.

Line 387: "higher volume" what does that mean? Presumably some connection to free energy?

Line 433: "novel perspective" I think the value is taking this perspective which is well known from 90s style statistical mechanics of learning and relating it to recent results in the machine learning literature.

Line 434: "important implications for the development of powerful optimization algorithms." I would like more details on this. I suspect that some of the references above could be examples of that and/or are consistent with that. But it isn't clear to me how one would "use" these results to, as a practical matter, develop improved algorithms.

Line 508: I don't think the insights are so immediately portable.

(Remarks on code availability)

Reviewer #2

(Remarks to the Author)

Summary

To explain why gradient descent (GD) can navigate complex loss landscapes and converge to solutions that generalize well, the authors develop a theoretical framework that models loss landscapes as multifractal. They demonstrate that their model provides a coherent explanation that links a set of geometrical signatures of loss landscapes—including multiscale features, highly degenerate minima, low barriers, and clustering—with optimization dynamics, including the EoS phenomenon, extended edge of chaos, and non-stationary anomalous diffusion.

By adapting fractional diffusion formalisms developed for non-equilibrium physical systems, they develop a fractional diffusion theory, which analytically illustrates how the dynamics of GD guide the optimizer toward large and smooth solution

spaces that house flatter minima. Their results indicate that the complexity inherent in loss landscapes does not impede optimization but facilitates the process.

Strength

The author proposed a unified theory framework that tries to explain different phenomena that are treated in an isolated way with separate models for each of them.

Their framework provides an alternative perspective for understanding deep learning, with possible implications for the development of powerful optimization algorithms.

Through developing the fractional diffusion theory, they elucidate that the non-stationary anomalous diffusive dynamics are functionally advantageous for deep learning.

Weakness

Analysis of real-world deep neural networks: the authors construct a minimal realization (2-D) of loss in Fig. 1 to illustrate how the multifractal loss landscape reconciles key properties found in real loss landscapes. The authors claim it is a minimal modeling approach, which is a common strategy in theoretical investigations of deep learning and grounded in the principles of statistical physics. While the illustration is straightforward, the example the author analyzed seems not really connected with practical deep learning, it would be more convincing if the authors can provide analysis of an real-world deep network as an example, such as the loss landscape of an image recognition model in <https://arxiv.org/abs/1712.09913>, which might attract interests of more broad audience. Similar as the author noted, “ the direct validation of multifractal loss structure in deep learning would require the development or extension of multifractal formalisms that are computationally efficient for high-dimensional surfaces”, there is still more work to do to further verify the analysis with practical problems.

Relationship with existing theoretical studies: The author claimed that according to their theory, the dynamics harness landscape-dependent annealing-like properties, steering the optimizer towards large and smooth solution spaces which house flatter minima. Given the strong correlation between solution flatness and generalization, they claimed their model effectively explains how the learning dynamics of GD can attain generalizing solutions in deep learning. It is not clear whether other training dynamics can also explain the effect and what makes the difference or advantage over other metrics.

Applications: the authors claimed that their finding not only has important implications for understanding deep learning but also extends potential applicability to other disciplines such as evolutionary biology and ecology, where optimization similarly unfolds on complex fitness landscapes, where optimization unfolds on complex landscapes. However, it is not clear how the analysis can benefit those areas.

(Remarks on code availability)

Reviewer #3

(Remarks to the Author)

This paper employs the multifractal loss landscape, and studies the behavior of gradient descent on it. The authors focus on the large learning rate gradient descent, where edge of stability and chaos appear, to establish a fractional diffusion theory and illustrate the advantages of the diffusion from GD and the complexity of the landscape in finding flatter minima.

Novelty:

The paper adopts a novel perspective, fractal loss, to study the common belief in machine learning application that large learning rates can find flatter minima. Specifically, the authors quantify the effect of the dynamics on the landscapes and the sharpness of the landscape using Holder exponent.

Questions:

1. This paper uses multifractal loss landscape. It is unclear how such landscape is related to the true landscapes in practice. For example, if assuming overparametrization, there will not be many spurious local minima and the global minima will be nonisolated. It would be better if explanations of these true landscapes could be provided from the perspective of this theory.
2. The anomalous diffusion of GD is a little vague to me, especially after seeing the figures in the supplementary materials. It would be better if more justification of the anomalous diffusion and the phase transition when t_w is small could be provided. Additionally, it is unclear to me whether it is the anomalous diffusion that helps or just any diffusion caused by large learning rates can help find flatter minima.
3. In the fractional diffusion theory, a continuous overdamped fractional Langevin equation is used. Under large learning rates, GD cannot be approximated by the continuous dynamics. It is still unclear how to interpret the theory even if the results from the dynamics match those of GD. More explanation should be provided.

Writing:

This paper is very technical and is a little hard to read probably due to limited information of the contexts. Maybe it will be better if more definitions and intuitive explanations about the concepts are provided, especially before introducing the results.

(Remarks on code availability)

Version 1:

Reviewer comments:

Reviewer #1

(Remarks to the Author)

This is a revision of a previous submission, which had interesting ideas but places them poorly in the existing literature on the statistical mechanics of learning. The authors substantially revised parts of the paper, following suggestions, and I think that the paper is a much more useful contribution now. Thanks to the authors for doing more than a pro-forma inclusion of suggested references, and instead going into them and placing their work much better with respect to those lines of work. I can recommend acceptance.

(Remarks on code availability)

Reviewer #3

(Remarks to the Author)

I would like to thank the authors for the detailed replies and the corresponding revision. The answers to the first and the third questions in my previous review are still unclear to me. More precisely,

For 1:

The authors added some discussions and experiments about the similarities between multifractal loss landscapes and the true landscapes (and some differences between them in supplementary materials). However, scientifically, it is still unclear how the theory can or cannot be generalized to the true landscapes.

For 3:

Regarding using gradient descent or gradient flow (or modified gradient flow) in theory, it's okay to use continuous dynamics in theory due to technical difficulties. However, phenomena such as edge of stability will not appear under gradient flow in machine learning models (see for example Cohen et al. "Gradient Descent on Neural Networks Typically Occurs at the Edge of Stability"). Large learning rate is still a necessary component in these observations and therefore is expected to show up in the theory if the authors would like to claim relevance. It seems there is still inconsistency in the theory between the (discrete) gradient descent with large learning rate and the (continuous) fractal diffusion theory, and also a gap between the theory in this paper and practice.

(Remarks on code availability)

Version 2:

Reviewer comments:

Reviewer #3

(Remarks to the Author)

I would like to thank the authors for the reply and revision. This paper studies the neural network landscape and training in a novel perspective but the previous version failed to build clear connections to real applications. The advantages and regimes of the theory in this paper are now much clearer, especially in its relationship to practical settings. My concerns are now fully addressed, and I would recommend acceptance.

(Remarks on code availability)

AUTHOR RESPONSE

Reviewer #1

This is a review of "Optimization on multifractal loss landscapes explains diverse range of geometrical and dynamical properties of deep learning."

The paper considers some of the recent work in machine learning having to do with learning with neural network models, and it presents a multifractal model that reproduces many of the properties. There has been a lot of work within machine learning on this general topic. Overall, the paper presents a number of interesting results that will probably be good for the machine learning community to know about.

We thank the reviewer for this positive appraisal of our work. Our detailed responses are given below, interleaved with the reviewer's comments (in blue).

They will probably not be surprising to the physics community. On the other hand, the paper presents the results by comparing to relatively straw men baselines (common in the machine literature, but naive from the statistical physics perspective). So, the paper actually reads more like a review of statistical physics type approaches, without any acknowledgement of many of the results in that statistical physics literature, instead only comparing to recent machine learning discussion of the topic.

Response: We thank the reviewer for pointing this out. We agree that it is important to reconcile the disconnect between machine learning and physics perspectives, and we greatly appreciate the reviewer's constructive comments that help our manuscript better achieve this. In the revised manuscript, we have taken multiple steps to strengthen baselines by acknowledging and discussing the relevant body of work about the statistical mechanics formulation of learning. Below we highlight each change in response to the specific comment raised by the reviewer.

With regard to the work in machine learning on this topic (which is the general framing of the paper), there is a lot of insight, there is also a large theory-practice disconnect. For example, most theory about SGD says very little about optimization dynamics, either since it is formulated for convex problems (until rather recently) or since it is on very idealized e.g. infinitely wide models (more recently). Some of the references below and other papers by those authors consider perspectives on optimization that are stronger baselines.

On the other hand, with regard to work in the statistical mechanics of learning, there is a large body of work that is ignored, and I should note that the work that is mentioned (e.g., the Bahri review, Ref 33) is not all of the statistical mechanics of learning, but only the part that is less relevant for the problems considered in that paper. Much more relevant is older work such as:

Seung et al., "Statistical mechanics of learning from examples," Physical Review A, 1992.
Haussler et al., "Rigorous learning curve bounds from statistical mechanics," Machine Learning, 1996.
Watkin et al., "The statistical mechanics of learning a rule," Rev. Mod. Phys., 1993.
Engel et al. "Statistical mechanics of learning," Cambridge University Press, 2001.
but there is also more recent work such as:

"Generalization Bounds using Lower Tail Exponents in Stochastic Optimizers," Hodgkinson et al., ICML22

"Taxonomizing local versus global structure in neural network loss landscapes," Yang et al., NeurIPS21

"Multiplicative noise and heavy tails in stochastic optimization," L. Hodgkinson et al., ICML21

"Implicit Self-Regularization in Deep Neural Networks: Evidence from Random Matrix Theory and Implications for Learning," C. H. Martin et al., JMLR21

Note that most of that work does not consider fractal/multifractal issues per se, so it doesn't undermine this paper. But that work does present results, either statistical mechanics formulations of the learning process or heavy tailed random matrix approaches or loss landscape or optimization dynamics

approaches that are much less straw men than much of the machine learning literature that is cited. There is also a body of work in multiple papers but summarized in the book:

Sornette. "Critical phenomena in natural sciences: chaos, fractals, selforganization and disorder: concepts and tools," Springer-Verlag, 2006.

some of which does consider fractality/multifractality.

Response: We thank the reviewer for highlighting the relevant literature on the statistical mechanics of learning. As suggested by the reviewer, we have revised the introduction and discussion sections to more clearly acknowledge the perspectives of this body of work as a stronger baseline. In particular, we have now emphasized studies from the statistical mechanics of learning literature that apply a dynamics approach to study generalization. We have added the following text to the introduction (L13, P. 2):

"A prevalent theoretical framework for understanding optimization processes in machine learning draws extensively on principles from statistical mechanics [6-17]. In particular, since the early studies on learning from examples, Langevin dynamics has been employed as a theoretical approach to analyze optimization algorithms and to calculate typical learning curves [6, 10-13]."

Additionally, we have elaborated on the statistical mechanics perspective of loss landscapes in the introduction (L43, P. 2), highlighting that this approach generally aims to quantify the global properties of the loss landscape, in contrast to some earlier machine learning perspectives that focus on local curvature:

"While these curvature metrics are mostly local, studies adopting a statistical mechanics perspective typically quantify more global structures in the loss landscape [9, 36-39]. Along this line, generalization has also been found to strongly correlate with the connectivity of solutions; the best generalization is associated with a rugged basin containing locally flat minima connected by highly degenerate low-loss paths [14]."

In the revised manuscript, we have also highlighted applications of the multifractal approach related to our work, including understanding the anomalous scaling of self-similar stochastic processes in physical systems and, the organization of neural network representations from the statistical mechanics of learning literature (L55, P. 3):

"Multifractality... has been applied to understand... the anomalous scaling behaviour of self-similar diffusion processes [44], the organization of internal representations in neural networks [9]..."

In summary, with these revisions and the additional changes detailed in the responses below, we have incorporated and discussed all the studies highlighted by the reviewer, as well as other relevant studies. These changes improve the positioning of our research within the broader context of state-of-the-art developments in machine learning and the statistical mechanics of learning.

I mention that since there are many comments throughout, e.g., starting with the abstract which says "... complexities of loss landscapes do not hinder optimization as conventionally thought," where the "conventionally" only refers to very recent work in machine learning that is ignorant about the older physics work. Similarly, edge of stability was well-known in the 90s statistical mechanics of learning, but ignored by the recent machine learning community. That's a statement about what the machine learning community chooses to pay attention to. But a paper like this should remedy that, especially since one of it's selling points is making connections with, e.g., non-equilibrium systems and other such complex physical systems and physical concepts, and being published in a non machine learning venue.

Response: As suggested by the reviewer, we have carefully revised our manuscript to ensure that relevant physics studies, such as those surrounding the statistical mechanics approach to learning, serve as stronger baselines. For instance, we have omitted comments like 'conventional wisdom' and

'conventionally,' which previously referred only to machine learning, to avoid referencing straw man perspectives. Furthermore, we agree that a goal of our work should be to improve the connection between machine learning and physical concepts. In the revised manuscript, to facilitate stronger connections between the perspectives of these fields, we have now interpreted the significance of our findings in relation to both the machine learning and statistical physics literature. For instance, we have situated our results about the edge of chaos in relation to its background in the statistical mechanics of learning as well as state-of-the-art methods in deep learning, including the addition of the following text in the discussion (L519, P. 17):

“This behavior enables convergence in distribution towards a solution space populated by well-connected flatter minima. Importantly, as the edge of chaos is extended—with its range increasing alongside the pointwise Hölder exponents that characterize multifractality—this enhanced generalization property does not require precise fine-tuning. It is worth noting that the concept of the edge of chaos has been proposed in previous theories of learning systems as an optimal setting for computation [59, 61]. In deep learning, the edge of chaos has been applied to layerwise signal propagation in deep networks. However, in contrast to our results, the order-to-chaos boundary in that context is non-extended, requiring stringent fine-tuning to achieve computational benefits such as preventing vanishing and exploding gradients [62, 63].”

There are multiple technical issues that would be good to clarify, even if it is in the supplementary material. One example is why work with the Frobenius norm of the Hessian, as opposed to top-k eigenvalue sum or trace. The explanation is just that it is a metric, but how does it compare to related metrics. This is relevant not just as a technical quibble, but also since there is a lot of work on Hessian properties "explaining" good generalization, which is at most a partial explanation since it is a very local metric.

Response: Following the reviewer's suggestion, we have now compared the Frobenius norm with common curvature metrics, including the spectral radius and top-k eigenvalue sum (trace) of the Hessian. Additionally, we have examined ϵ -sharpness, a “non-local” measure of sharpness, to demonstrate that both local and non-local measures agree well in their ranking of the curvature of minima within the multifractal loss landscape. These results indicate that our analysis, under the common assumption of the flat minima conjecture, can explain generalization regardless of the specific curvature metrics. To summarize these new results, we have added a new section to the Supplementary Information (Sec. 3 on P. 3) and a new figure (Fig. S3).

“Since various curvature metrics are often correlated with generalization, we compare the Frobenius norm to other common curvature metrics to assess their consistency in the context of the multifractal loss landscape. In particular, we evaluate the spectral radius of the Hessian λ_{\max} [15], the trace of the Hessian $\text{Tr}(H)$ and the ϵ -sharpness [16], calculated as:

$$\frac{\max_{\theta' \in B_\epsilon^\infty} (L(\theta') - L(\theta))}{1 + L(\theta)},$$

where B_ϵ^∞ is the ϵ -ball in the l^∞ norm. This modification of the sharpness metric in [16], similar to that in [17], adopts the l^∞ norm to consider points within a square of length 2ϵ centered at θ on the Cartesian grid. We also note that this measure of curvature is non-local, contrasting with other local, Hessian-based metrics. Figure S3 shows that all metrics consistently identify flatter minima within the multifractal landscape. To quantify this consistency, we also measure the Spearman rank correlation coefficients, obtaining the values of 0.995, 0.993 and 0.828 for the spectral radius, trace and ϵ -sharpness (with $\epsilon = 5$), respectively. These high correlation values confirm that smoother regions of the multifractal landscape consistently house flatter minima across all examined curvature metrics.”

Additionally, as mentioned earlier in response to the first comment, we have now incorporated context about the connection between generalization and connectivity, a more global property compared to curvature. To relate our results to this context, we have added the following text on L170, P. 6:

“Moreover, the clustering of highly degenerate solutions within smoother basins aligns with theoretical and empirical characterizations of the global structure of loss landscapes from a statistical mechanics perspective [9, 36-39]. A major implication of these studies is that the solution space could be described as a manifold of well-connected, non-isolated minima that is highly navigable for an optimizer. Importantly, such connectivity among solutions has been found to strongly correlate with generalization [14].”

We have also rewritten the final sentence in the results section to explicitly highlight that our analysis explains generalization from both the perspectives of achieving (locally) flatter solutions and (globally) well-connected solutions (L461, P. 16):

“Given that generalization has been strongly correlated with both local flatness [24, 32-35] and global connectivity [14], our model effectively explains how the learning dynamics of GD can attain generalizing solutions in deep learning.”

We have also added a sentence to the discussion (L484, P. 16) proposing that the multifractal structure can account for observed correlations between generalization and both local and global properties of the loss landscape:

“This predicted correlation between smoothness and flatness may explain why global connectivity and local curvature are both associated with generalization [14].”

The figures were reasonably convincing that the suggested model explained many of the qualitative properties it aims to explain. It would be good, however, even if in an appendix to have more details about the models considered, variants to the model trained to different types of data, etc., i.e., "controls" (in science) or "ablations" in machine learning.

Response: Following the reviewer’s advice, we have performed additional control studies involving variants of our multifractal landscape model. Specifically, we have characterized the properties of GD dynamics on monofractal and random landscapes, as well as loss landscape models previously considered in the machine learning literature, including two-scale (Ref. 22) and highly degenerate (Ref. 45) structures. These new control analyses indicate that the multifractal loss landscape is particularly robust in explaining the dynamical properties of GD optimization, including the extended edge of chaos and anomalous diffusion. To present these new results, we have rewritten Sec. 10 in Supplementary Information (P. 17) which includes the following text, and reorganized Fig. S9 to facilitate a clearer comparison between these variants and to include new results for the time averaged mean squared displacement in these variants.

“As a control study, we investigate the dynamics of GD on various modifications of the loss landscape model. We note that there have been efforts to relate the structure of the loss landscape to data characteristics and network architecture. For example, visualizations have shown that deeper networks exhibit more “chaotic” loss landscapes [20], and multiscale training data can cause multiscale loss landscapes [22]. However, a precise relationship among these factors remains unclear. Thus, we consider a range of variants to the loss landscape, including monofractal, random, two-scale, and highly degenerate structures...”

We have also re-organized the results of the control study in Sec. 11 in Supplementary Information. These results confirm the robustness of our analysis against variations in the pointwise Hölder exponent of the multifractal loss landscape. To enable clearer comparisons between the variants, we have combined several figures (Fig. S6-S10 previously) into a single figure (Fig. S10 now).

At line 180, it is said (correctly) that SGD analysis usually makes assumptions about L-smoothness, etc. I said above the this reads more like a review than a new paper since, e.g., that analysis is about convex problems, etc., which clearly are not applicable to the learning process. (Admittedly, machine learning papers before a few years ago did make such assumptions, but that doesn't mean that a paper like this should use that as a baseline. The Hodgkinson paper above considers a much more realistic baseline. To my mind, this contrasts with much of the edge of chaos type discussion which (even though not well placed in the physics literature, which knew about it long ago) does make what to my knowledge are new connections with fractality, in the machine learning community at least.

Response: As suggested by the reviewer, we have rewritten this paragraph in the revised manuscript to briefly contextualize the EoS phenomenon before addressing the main point in the paragraph. In the revised manuscript, we have now noted that the EoS phenomenon, although challenging classical ideas of optimization, does not preclude learning in state-of-the-art optimization theories with more realistic baselines, including the Hodgkinson paper. The following text has been added (L207, P.9):

“Contrary to previous wisdom for efficient optimization based on assumptions of L-smoothness regularity [30, 31], recent studies on the EoS phenomenon have shown that larger learning rates η marginally violating the stability condition (i.e., $\eta > 2/\lambda_1$, where λ_1 is the leading Hessian eigenvalue) lead to superior performance [29]. We note that in recent optimization theories with more realistic regularity assumptions, such as Lipschitz continuous and bounded loss [26], violating the stability condition does not preclude learning. Since the EoS phenomenon is a characteristic feature of neural network training with GD, we demonstrate here that it naturally arises in our model.”

Additionally, as demonstrated in our response above, we have improved the placement of our results about the edge of chaos in the physics literature. Specifically, we have highlighted the novel connection between the extended edge of chaos and multifractality revealed by our study.

Line 256: I'm not sure about the explanation for curse of dimensionality. Also, the mention of "realistic examples" leads me to want more details on the details of the models presented in the figures.

Response: To avoid any confusion, we have rewritten this point about the curse of dimensionality. In the revised manuscript, we have now clarified that the apparent absence of saturation in previous experiments demonstrating anomalous diffusion in real neural network training is simply due to insufficient lag times. To justify this, we have added a new reference containing similar MSD calculations (L293, P. 11), showing saturation when much longer lag times are used:

“This non-stationary anomalous diffusion is similar to that recently observed in realistic training scenarios [22]. A minor discrepancy is the apparent absence of saturation in these examples. However, some studies considering long timescales demonstrate the presence of saturation [24].”

In the revised manuscript, we have also clarified that the mention of “realistic examples” refers to previous results involving real neural network training. Nonetheless, we have performed new analysis to connect our multifractal theory more closely to these realistic examples. In particular, we have demonstrated that the true loss landscapes of deep neural networks exhibit multifractal structure using the visualization method of Ref. 55. To summarize these new results, we have added a new figure (Fig. 3), a new section on P. 7, a new section in the Methods on P. 21 to explain the analysis, and a new supplementary section (Sec. 4 on P. 4 of the Supplementary Information) with further investigation of the robustness of the results to changes in dataset, optimizer and architecture.

I think that the value of the paper is putting a lot of work in a common setup. But I think that its effectiveness is hindered by not placing it well in the context of state of the art methods in the machine learning literature on optimization dynamics, loss landscapes, random matrix theory, etc., as well as state of the art methods from 90s style statistical physics of learning.

Response: We thank the reviewer for appreciating the value of the paper in offering a coherent explanation for a great variety of deep learning phenomena. To improve its effectiveness, we have now improved the placement of our work within the context of state-of-the-art methods in machine learning and statistical physics. In particular, we have now considered the literature on optimization dynamics, including the Langevin dynamics approach extending from the statistical mechanics of learning, as well as recent machine learning theories with heavy-tailed stochastic behaviour. We have highlighted connections to heavy-tailed random matrix theory for the latter. In addition to other changes throughout the revised manuscript as shown in earlier responses, we have also rewritten a paragraph of the discussion (L529, P. 17) to place our work within this context:

“There is a large body of related work applying statistical mechanics approaches to understand optimization processes [6-17]. An important idea originating from the statistical mechanics formulation of learning is that the dynamical behavior of learning algorithms could be studied to understand the energy landscape itself [7, 16]. In particular, Langevin dynamics as a theoretical framework has extended from early studies within this context [6] to more recent investigations of SGD [18, 19]. Since empirical evidence of more complex stochastic behavior has emerged, such as anomalous diffusion [22, 23] and heavy-tailed statistics [15], optimization dynamics with non-Gaussian heavy-tailed noise [21] and multiplicative noise [27] have been considered. The advantages of these dynamics for generalization are understood through heavy-tailed random matrix theory, which is deep-seated in statistical physics [15].”

Regarding the loss landscape, we have strengthened the baseline for the relationship between generalization and geometrical structure by incorporating the statistical mechanics perspective that connects generalization to global connectivity, as mentioned in the previous responses. Correspondingly, we have emphasized that our model explains generalization from both the perspective of flatter minima and better connectivity.

Line 387: "higher volume" what does that mean? Presumably some connection to free energy?

Response: In the revised manuscript, we have rewritten the sentence on L422 P. 15 to clarify what we mean by a higher-volume solution space. Specifically, we have elaborated that this refers to the size of the connected set where the potential energy is low, which has been formalized as the volume ϵ -flatness in Ref. 56 based on the original notion of flatness in Ref. 32. In this context, a larger volume entails a greater capacity for housing low-loss, degenerate solutions:

“This preference translates into a prediction for higher-volume solution spaces in the sense that the connected set, whose potential is similar to the minimum, is larger; such a solution space inherently possess a greater capacity for hosting a larger cluster of highly-degenerate solutions.”

Line 433: "novel perspective" I think the value is taking this perspective which is well known from 90s style statistical mechanics of learning and relating it to recent results in the machine learning literature.

Response: In the revised manuscript, we have rewritten the sentence on L473, P. 16 in light of the improved connections to the statistical mechanics literature:

“Our framework thus provides a unifying, non-equilibrium statistical mechanics approach for understanding deep learning.”

Line 434: "important implications for the development of powerful optimization algorithms." I would like more details on this. I suspect that some of the references above could be examples of that and/or are consistent with that. But it isn't clear to me how one would "use" these results to, as a practical matter, develop improved algorithms.

Response: In the revised manuscript, we have provided further details on the implications of our results for the development of improved algorithms. As a proof of concept, we have constructed and tested a simple optimization algorithm that utilizes a primary insight, which is the functional benefit of landscape-dependent anomalous diffusion, to target flatter minima. Specifically, we have applied heavy-tailed fluctuations with an adaptive tail index to promote exploration, consistent with references above about heavy-tailed dynamics, while balancing with exploitation through slower diffusion in flatter regions. This heuristic can be extended to other geometrical properties of the loss landscape (e.g., roughness), but we agree that future investigation is needed to apply these concepts in a practical and efficient manner to improve algorithms. To summarize these new results, we have added a new section in the Supplementary Information (Sec. 13 on P. 23) and a new figure (Fig. S13).

Line 508: I don't think the insights are so immediately portable.

Response: We agree that the claim was presumptive. In the revised manuscript, we have replaced it with a statement that future work regarding the dependence of SGD noise on multifractal roughness of the loss landscape would be of interest (L557, P. 18):

“The insights gained may also be valuable for future investigation into whether SGD noise is dependent on the roughness of a multifractal loss landscape, providing a more complete picture of neural network training dynamics.”

Reviewer #2

Summary

To explain why gradient descent (GD) can navigate complex loss landscapes and converge to solutions that generalize well, the authors develop a theoretical framework that models loss landscapes as multifractal. They demonstrate that their model provides a coherent explanation that links a set of geometrical signatures of loss landscapes—including multiscale features, highly degenerate minima, low barriers, and clustering—with optimization dynamics, including the EoS phenomenon, extended edge of chaos, and non-stationary anomalous diffusion.

By adapting fractional diffusion formalisms developed for non-equilibrium physical systems, they develop a fractional diffusion theory, which analytically illustrates how the dynamics of GD guide the optimizer toward large and smooth solution spaces that house flatter minima. Their results indicate that the complexity inherent in loss landscapes does not impede optimization but facilitates the process.

Strength

The author proposed a unified theory framework that tries to explain different phenomena that are treated in an isolated way with separate models for each of them. Their framework provides an alternative perspective for understanding deep learning, with possible implications for the development of powerful optimization algorithms. Through developing the fractional diffusion theory, they elucidate that the non-stationary anomalous diffusive dynamics are functionally advantageous for deep learning.

Response: We thank the reviewer for this positive appraisal of our work and constructive feedback. Our detailed responses are given below, interleaved with the reviewer's comments (in blue).

Weakness

Analysis of real-world deep neural networks: the authors construct a minimal realization (2-D) of loss in Fig. 1 to illustrate how the multifractal loss landscape reconciles key properties found in real loss landscapes. The authors claim it is a minimal modeling approach, which is a common strategy in theoretical investigations of deep learning and grounded in the principles of statistical physics. While the illustration is straightforward, the example the author analyzed seems not really connected with practical deep learning, it would be more convincing if the authors can provide analysis of an real-world deep network as an example, such as the loss landscape of an image recognition model in <https://arxiv.org/abs/1712.09913>, which might attract interests of more broad audience. Similar as the author noted, “the direct validation of multifractal loss structure in deep learning would require the development or extension of multifractal formalisms that are computationally efficient for high-dimensional surfaces”, there is still more work to do to further verify the analysis with practical problems.

Response: We appreciate the constructive feedback very much. To better connect the multifractal loss landscape model to practical deep learning and thus attract the interest of a broader audience, we have analysed the loss landscapes of realistic deep neural networks. Specifically, by using the visualization method of Ref. 55 (<https://arxiv.org/abs/1712.09913>) as suggested by the reviewer, we have characterized the loss landscape in practical deep learning problems. Our new analysis reveals that the two-dimensional section of the loss landscape exhibits a multifractal structure with a diverse multifractal spectrum. We have also demonstrated that these results are robust to changes in training data and optimizer, and we have further characterized fractal clustering in these surfaces as done in our theoretical model. Notably, the multifractal spectrum broadens further during training, indicating the optimizer navigates towards smoother regions of the loss landscape, consistent with our theoretical predictions. To present these new results, we have added a new figure (Fig. 3) and a new section on P. 7 with the following text. Correspondingly, we have added a new section in the Methods to detail the analysis (P.

21), and a new section in the Supplementary Information (Sec. 4 on P. 4) to present results demonstrating robustness and further characterizations.

“We now empirically relate the multifractal model to the loss landscape of realistic deep neural networks. In particular, by applying the filter normalization method in [55], we visualize and characterize two-dimensional sections of the loss landscape centered on the location of the optimizer at different epochs during the training of a VGG-16 network on the CIFAR-10 dataset (see Methods for further details on the experimental configuration and other types of networks and data). The orientation of the two-dimensional sections is defined by filter-normalized random directions. This method is widely used because it enables meaningful comparisons of loss structure by preventing distortions caused by the scale invariance of networks with ReLU non-linearities [55–58]. Figure 3a shows the loss surface after 20 epochs. The surface exhibits a low-loss basin at the optimizer position that appears smoother than surrounding regions of higher loss. To quantify these variations in roughness across the surface, we calculate the multifractal singularity spectrum, which shows a continuous set of scaling exponents (Fig. 3b), indicating the presence of multifractal structure. In addition, we find that the singularity spectrum broadens towards larger exponents α during training. This is because the loss visualization is centered at the optimizer, and the optimizer navigates towards a smoother basin. This finding is further supported by estimations of the pointwise Hölder exponent across the two-dimensional visualizations, which quantifies the precise spatial variation in roughness (see Supplementary Sec. 4). Further characterizations of roughness do not change significantly after 20 epochs, which approximately coincides with when accuracy reaches a maximum and plateaus. Correspondingly, as our fractional diffusion theory will later demonstrate, GD in a multifractal loss landscape exhibits a preference for smoothness that benefits generalization. In Supplementary Sec. 4, we show these results are robust to changes in training data and optimizer, and we further characterize fractal clustering in these surfaces as done in our model.”

Relationship with existing theoretical studies: The author claimed that according to their theory, the dynamics harness landscape-dependent annealing-like properties, steering the optimizer towards large and smooth solution spaces which house flatter minima. Given the strong correlation between solution flatness and generalization, they claimed their model effectively explains how the learning dynamics of GD can attain generalizing solutions in deep learning. It is not clear whether other training dynamics can also explain the effect and what makes the difference or advantage over other metrics.

Response: As suggested by the reviewer, we have now experimentally clarified the differences between the GD training dynamics in our analysis and previously suggested training dynamics. To achieve this, we have empirically compared our proposed GD dynamics (i.e., fractional Langevin equation) with multiple models of SGD training dynamics (i.e., Gaussian, Lévy and multiplicative noises). These new results suggest that the learning dynamics of GD emerging in a multifractal landscape exhibits a unique mechanism: landscape-dependent anomalous diffusion. This mechanism enables GD to target generalizing solutions in a manner resembling an annealing-like strategy. To summarize these new results, we have added a new supplementary section (Sec. 12 on P. 19) and two new figures (Fig. S11 and Fig. S12). For instance, we have added the following text in the supplementary section (L557, P. 22):

“To help illustrate the differences between the current study and prior studies, in this section we conduct a variation of the numerical experiment as proposed in [46]. Specifically, we simulate various training dynamics on a non-convex toy loss function with many minima...”

Regarding metrics for generalization, we have now compared various curvature metrics, including local and non-local ones, to demonstrate that all metrics consistently identify flatter minima in the multifractal loss landscape. We have also recognized that solution connectivity, a more global property than curvature, has been found to correlate with generalization. To integrate this context into the manuscript, we have added the following text to the introduction (L43, P. 2):

“While these curvature metrics are mostly local, studies adopting a statistical mechanics perspective typically quantify more global structures in the loss landscape [9, 36-39]. Along this line, generalization has also been found to strongly correlate with the connectivity of solutions; the best generalization is associated with a rugged basin containing locally flat minima connected by highly degenerate low-loss paths [14].”

Accordingly, we have rewritten multiple sentences throughout the revised manuscript to emphasize that our analysis explains generalization from both the perspectives of achieving (locally) flatter solutions and (globally) well-connected solutions. For example, we have rewritten the final sentence in the results section (L461, P. 16):

“Given that generalization has been strongly correlated with both local flatness [24, 32-35] and global connectivity [14], our model effectively explains how the learning dynamics of GD can attain generalizing solutions in deep learning.”

Applications: the authors claimed that their finding not only has important implications for understanding deep learning but also extends potential applicability to other disciplines such as evolutionary biology and ecology, where optimization similarly unfolds on complex fitness landscapes, where optimization unfolds on complex landscapes. However, it is not clear how the analysis can benefit those areas.

Response: In the revised manuscript, we have further elaborated on how our analysis might extend its applicability to related fields. Notably, since the literature on biological fitness landscapes reveals similar fractal properties, our analysis could have significant implications for understanding optimization processes in such landscapes. To clarify this point, we have added the following text to the discussion on L567, Page 18:

“For example, statistical analyses of biological fitness landscapes have similarly connected spatial correlations in the landscape to fractality [48]. Our analysis could provide a framework for understanding how biological fitness landscapes, despite their high dimension and extreme ruggedness, remain easily navigable [49]. Consistent with our findings, the complexity of such landscapes may not impede navigation or optimization processes, but may instead facilitate them.”

Reviewer #3

This paper employs the multifractal loss landscape, and studies the behavior of gradient descent on it. The authors focus on the large learning rate gradient descent, where edge of stability and chaos appear, to establish a fractional diffusion theory and illustrate the advantages of the diffusion from GD and the complexity of the landscape in finding flatter minima.

Novelty:

The paper adopts a novel perspective, fractal loss, to study the common belief in machine learning application that large learning rates can find flatter minima. Specifically, the authors quantify the effect of the dynamics on the landscapes and the sharpness of the landscape using Holder exponent.

Response: We thank the reviewer for this positive appraisal of our work and constructive feedback. Our detailed responses are given below, interleaved with the reviewer's comments (in blue).

Questions:

1. This paper uses multifractal loss landscape. It is unclear how such landscape is related to the true landscapes in practice. For example, if assuming overparametrization, there will not be many spurious local minima and the global minima will be nonisolated. It would be better if explanations of these true landscapes could be provided from the perspective of this theory.

Response: As suggested by the reviewer, we have now related practical loss landscapes to the perspective of our theory by characterizing the actual loss landscapes of deep neural networks. This was achieved using the visualization method from Ref. 55 (<https://arxiv.org/abs/1712.09913>) and employing the same analytical approach as in our mathematical model. Our results confirm that true loss landscapes exhibit multifractal structure consistent with our mathematical model, as characterized by a multifractal singularity spectrum that broadens during training. To present these new results, we have added a new figure (Fig. 3) and a new section on P. 7 with the following text. Correspondingly, we have added a new section in the Methods to explain our analysis (P. 21) and a new section in the Supplementary Information (Sec. 4 on P. 4) that provides further characterization of the multifractal structure and additional investigation of robustness of the results to changes in dataset, optimizer and architecture.

“We now empirically relate the multifractal model to the loss landscape of realistic deep neural networks. In particular, by applying the filter normalization method in [55], we visualize and characterize two-dimensional sections of the loss landscape centered on the location of the optimizer at different epochs during the training of a VGG-16 network on the CIFAR-10 dataset...”

Additionally, in the revised manuscript, we have further explored realistic aspects of true loss landscapes from the perspective of our theory. Specifically, we have empirically observed that local minima in the multifractal landscape are confined within a band of low loss values, aligning with previous theoretical results that suggest the absence of spurious local minima. To summarize this finding, we have added a new paragraph (L60, P. 3) and a new figure (Fig. S2) to Section 2 of the Supplementary Information. Regarding non-isolated global minima, we interpret this property as indicative of the connectivity of solutions through highly degenerate paths; for instance, Draxler et al. (2018) empirically observed this, and Yang et al. (2021) linked the connectivity of solutions in rugged basins to generalization. In the revised manuscript, we have revised several sentences in the Results section (L170, Page 6) to clarify that the multifractal landscape exhibits this feature:

“Moreover, the clustering of highly degenerate solutions within smoother basins aligns with theoretical and empirical characterizations of the global structure of loss landscapes from a statistical mechanics perspective [9, 36-39]. A major implication of these studies is that the solution space could be described

as a manifold of well-connected, non-isolated minima that is highly navigable for an optimizer. Importantly, such connectivity among solutions has been found to strongly correlate with generalization [14].”

2. The anomalous diffusion of GD is a little vague to me, especially after seeing the figures in the supplementary materials. It would be better if more justification of the anomalous diffusion and the phase transition when t_w is small could be provided. Additionally, it is unclear to me whether it is the anomalous diffusion that helps or just any diffusion caused by large learning rates can help find flatter minima.

Response: We appreciate the constructive feedback. We have now added a new section in the Supplementary Information (Sec. 6 on P. 9) to justify the anomalous diffusion. Specifically, in the revised manuscript, we have provided additional explanation on how the features of the time-averaged mean squared displacement in the figures physically relate to specific aspects of GD interacting with a multifractal landscape. For example, we have explained that the transition from sub-diffusion to super-diffusion at small waiting times occurs due to the positive correlation of steps caused by a bias in the landscape (L260, P. 10):

“Similarly, the transition to transient super-diffusion at short waiting times t_w arises from correlated steps. Specifically, the bias of the tilted washboard potential (or more general loss structure) drives the optimizer in a particular direction, resulting in the persistence (positive correlation) of steps.”

In the revised manuscript, we have included additional analysis that underscores the critical role of anomalous diffusion—as opposed to normal diffusion—in facilitating the discovery of flatter minima. This analysis is supported by a new set of numerical experiments that compare the training dynamics in our study with those previously proposed. Our findings demonstrate that GD in a multifractal loss landscape utilizes a distinctive mechanism to find generalizing solutions, specifically through its landscape-dependent anomalous diffusion that mimics an annealing-like strategy. To summarize these new results, we have added a new section in Supplementary Information (Sec. 12 on P. 19) and a new figure (Fig. S11):

“Fig. S11 further demonstrates that anomalous diffusion, as opposed to normal diffusion, is central to facilitating this annealing-like behaviour of GD in a multifractal loss landscape...”

Accordingly, we have rewritten a sentence (L450, P. 16) in the main text to clarify the importance of anomalous diffusion:

“For deeper insights, consider the large t limit where equation (17) behaves as $P(t) \sim 1/t^{s_0^\alpha}$, where $\alpha = 2 - 2H$ is the diffusion exponent. The power-law exponent reveals the important role of anomalous diffusion in the exploitation ability of GD, since the slower diffusion rate (smaller α) in smoother basins result in longer escape times (see Supplementary Sec. 12 for further numerical evidence).”

3. In the fractional diffusion theory, a continuous overdamped fractional Langevin equation is used. Under large learning rates, GD cannot be approximated by the continuous dynamics. It is still unclear how to interpret the theory even if the results from the dynamics match those of GD. More explanation should be provided.

Response: As suggested by the reviewer, we have now provided further explanation of how to interpret the theory in light of the validity of continuous-time limits for approximating iterative gradient-based algorithms. In the revised manuscript, we have clarified that it is a minimal operational theory capturing empirical findings, which is a common approach in theoretical studies of deep learning (see e.g., Refs. 6, 15, 28, 54). We have also clarified that we use the continuous-time setting because it is analytically tractable based on fractional calculus formalisms suitable for describing anomalous diffusion induced

by temporally correlated dynamics. This approach also facilitates practical insights that would be otherwise more challenging in the discrete case. Indeed, although it is an approximation, the continuous case quantitatively captures the main dynamics of our model. Drawing from previous deep learning theories that were initially formulated in continuous time and later successfully translated to discrete time (Refs. 25, 26), we agree that the future development of the fractional diffusion theory in a discrete-time context represents an important and non-trivial mathematical task. We have added a paragraph (L410, P. 15) at the end of Sec. 8 of the Supplementary Information to explain this:

“We now clarify how to interpret the fractional diffusion theory in light of recent results regarding the use of continuous-time limits for modelling iterative gradient-based methods [32-35]. The main arguments about using continuous-time limits are, first, in approximations of GD as modified differential equations, higher-order error terms grow when the learning rate is sufficiently large [22] and, second, in approximations of SGD as stochastic differential equations, the noise term vanishes in the continuous limit [36]. Despite these challenges, the continuous-time framework remains valuable for deriving theoretical results that are adaptable to discrete settings. For example, theories about generalization bounds based on heavy-tailed dynamics were initially formulated in continuous-time [37] and were later successfully translated to discrete time applications [38]. Similarly, the future development of the fractional diffusion theory in a discrete-time context represents a non-trivial mathematical task. Our current continuous-time theory should be interpreted as a minimal operational theory because it explains: first, how the interplay between GD dynamics and loss structure induces complex dynamics, such as anomalous diffusion; second, how such complex dynamics are advantageous for generalization. Fractional calculus, specifically the fractional Langevin equation, offers a suitable mathematical framework to describe anomalous diffusion induced by temporally correlated dynamics.”

Writing:

This paper is very technical and is a little hard to read probably due to limited information of the contexts. Maybe it will be better if more definitions and intuitive explanations about the concepts are provided, especially before introducing the results.

Response: We agree with the reviewer that making the paper accessible to a broad audience is essential yet challenging due to the interdisciplinary nature of our study. In the revised manuscript, we have provided additional context, such as previous and contemporary applications of statistical mechanics approaches to understanding learning processes. We have also revised and added sentences to clarify concepts before presenting results. For instance, prior to discussing the outcomes of multifractal analyses, we have expanded on the definition of the multifractal singularity spectrum (L134, P. 5) and added intuitive explain for the equation (4) (L96, P.4)

“Thus, $f(\alpha)$ represents the space-filling capacity of the set of points that have a pointwise Hölder exponent value of α .”

“This covariance captures the spatial correlations in the structure of B_H that gives rise to fractal properties.”

Furthermore, we have added intuitive explanations to concepts—such as non-stationary anomalous diffusion, the time-averaged mean squared displacement (Section 6 of the Supplementary Information) and their connections to learning—in various sections of the Supplementary Information. In general, to minimise the technicality of the main text, we have kept technical aspects of our study (e.g., mathematical derivations) to the Supplementary Information with references made in the main text to indicate their availability.

AUTHOR RESPONSE

Reviewer #1

This is a revision of a previous submission, which had interesting ideas but places them poorly in the existing literature on the statistical mechanics of learning. The authors substantially revised parts of the paper, following suggestions, and I think that the paper is a much more useful contribution now. Thanks to the authors for doing more than a pro-forma inclusion of suggested references, and instead going into them and placing their work much better with respect to those lines of work. I can recommend acceptance.

We thank the reviewer for their time and constructive feedback that has contributed to the improvement of our manuscript.

Reviewer #3

I would like to thank the authors for the detailed replies and the corresponding revision. The answers to the first and the third questions in my previous review are still unclear to me. More precisely,

We thank the reviewer for further comments. Our detailed responses are given below, interleaved with the reviewer's comments (in blue).

For 1: The authors added some discussions and experiments about the similarities between multifractal loss landscapes and the true landscapes (and some differences between them in supplementary materials). However, scientifically, it is still unclear how the theory can or cannot be generalized to the true landscapes.

Response: As suggested by the reviewer, in the revised manuscript, we have now clearly illustrated the general applicability of our theory to true landscapes. Since true landscapes are influenced by many factors in a highly complex manner, including data and architectural influences, previous investigations into the effects of these factors on the landscape's geometrical structure have primarily relied on empirical approaches (Li et al., 2018; Draxler et al., 2018; Verpoort et al., 2020; Xie et al., 2024). Similarly, we have presented empirical findings that support our theory's generalizability to true landscapes when they exhibit multifractal characteristics, as indicated by a broad multifractal singularity spectrum, $f(\alpha)$, but not otherwise. We have now demonstrated this from two major perspectives: first, the geometrical structure of true landscapes, and second, the resulting training dynamics of GD.

First, to determine whether our theory generalizes to true landscapes in terms of their geometrical structure, we have demonstrated that the multifractal spectrum, $f(\alpha)$, serves as an effective theoretical index for capturing multifractal characteristics in true landscapes. To this end, we have examined deep neural networks (including VGG networks) whose true landscapes—specifically two-dimensional sections visualized using state-of-the-art methods (Li et al., 2018)—exhibit multifractal characteristics throughout all training stages and across multiple changes to training configuration. We have revealed that the multifractal spectra of these true landscapes are consistently broad, effectively capturing their multifractal nature (i.e., a wide range of pointwise Hölder exponents). To further establish the utility of the multifractal spectrum as an index, particularly for distinguishing between multifractal and non-multifractal characteristics in true landscapes, we have also revealed deep neural networks (including VGG networks with dropout layers ablated and ResNet networks) with true landscapes that are non-multifractal. In these cases, the shape of the multifractal spectrum is distinctly different due to spurious fitting results, signifying non-fractal geometry. As we further clarify below, our theoretical analysis of training dynamics only generalizes to true landscapes with multifractal characteristics, as indicated by the verifiable criterion of a broad multifractal spectrum, but not otherwise. In the revised manuscript, we have rewritten Supplementary Sec. 4 (L98, P. 4 to L165, P. 7 of the Supplementary Information) and added a new figure (Fig. S6) to clarify the multifractal spectrum as an effective indicator of multifractal structure in true landscapes.

Second, in the revised manuscript, we have further demonstrated that our theory generalizes to true landscapes in terms of training dynamics. Specifically, we have conducted extensive real-network experiments with systematic changes to training configuration (e.g., learning rate and dropout rate) to characterize training dynamics in true landscapes with and without multifractality, using a broad multifractal spectrum as the criterion for multifractality. As a result, we have unveiled non-stationary anomalous diffusion in true landscapes with multifractal characteristics, as explained by our theory. Conversely, these complex diffusive dynamics do not occur when the landscape is not multifractal. Taken together, these new experimental results confirm that our theory generalizes to true landscapes that meet verifiable criteria based on a broad multifractal spectrum. In the revised manuscript, we have extended Supplementary Sec. 4 (L175, P. 7 to L224, P. 9 of the Supplementary Information) and added new figures (Fig. S8 and Fig. S9) to detail these new results.

In summary, our theory provides the first coherent explanation that links a diverse set of geometrical signatures of loss landscapes to various optimization dynamics. Our work thus paves the way for further investigations into the relationships among key factors, such as data and architectural influences, the geometry of the loss landscape, and gradient-based training dynamics. Moreover, our findings suggest that, to help uncover these complex relationships, future research in the refinement of tools for multifractal analysis is important. In addition to the changes outlined above, we have rewritten a paragraph in the main text (L299, P. 11) to explicitly clarify the conditions under which our analysis can be generalized to true landscapes:

“We now describe the conditions under which our analysis is applicable to real networks. Theoretically, the loss function must exhibit the multifractal geometric structure characteristic of our landscape model. Underlying our landscape model are the following assumptions... Because the true loss landscape is influenced by many factors, such as data properties and architectural characteristics, in a highly complex manner (Xie et al., 2024), it is mathematically challenging to determine the contributions of these factors to the landscape’s geometrical structure. Consequently, previous investigations into these effects have relied on empirical approaches (Verpoort et al., 2020; Draxler et al., 2018; Li et al., 2018; Xie et al., 2024) showing, for instance, that the complexity of the loss landscape increases with task difficulty (Xie et al., 2024) but decreases with network width (Li et al., 2018). Similarly, our empirical findings corroborate the general applicability of our theoretical analysis when the loss landscape, specifically a two-dimensional section of it (Li et al., 2018), exhibits multifractal characteristics, as indicated by a broad multifractal singularity spectrum, but not applicable otherwise; see Supplementary Sec. 4 for extensive real-network experiments supporting this.”

For 3: Regarding using gradient descent or gradient flow (or modified gradient flow) in theory, it's okay to use continuous dynamics in theory due to technical difficulties. However, phenomena such as edge of stability will not appear under gradient flow in machine learning models (see for example Cohen et al. "Gradient Descent on Neural Networks Typically Occurs at the Edge of Stability"). Large learning rate is still a necessary component in these observations and therefore is expected to show up in the theory if the authors would like to claim relevance. It seems there is still inconsistency in the theory between the (discrete) gradient descent with large learning rate and the (continuous) fractal diffusion theory, and also a gap between the theory in this paper and practice.

Response: We thank the reviewer for raising this point. Inspired by this comment, we have performed new mathematical analyses demonstrating that the continuous-time fractional Langevin equation can approximate discrete-time gradient descent with a large learning rate. Our analytical findings are supported by comprehensive simulations, highlighting the consistency between our theoretical predictions and empirical observations. Our new analysis is based on the following major steps:

First, to facilitate mathematical analysis, we have considered a multiscale loss function composed of n power-law distributed scales. In the infinite- n limit, this loss function approaches a scale-free function that

exactly represents the landscape model of our multifractal theory locally. We have provided numerical evidence to demonstrate this limit in a new figure, Fig. S11(a-b) of the Supplementary Information, which is shown below. Using this approach, we have explicitly derived the relationship between the pointwise Hölder exponent of the loss landscape and the anomalous diffusive exponent of GD dynamics, as discussed in the main text.

Second, motivated by recent results on the chaotic dynamics of GD with large learning rates (Kong & Tao, 2020), we have approximated the gradient of small-scale components of the loss function as fractional Gaussian noise in the large learning-rate regime. We have assumed the remaining large-scale components of the loss function represent a potential term in the dynamical equation. Intuitively, the steps parallel the decomposition of the loss landscape to model large learning-rate GD as described in Kong & Tao (2020). Using a conventional statistical mechanics approach involving the fluctuation-dissipation theorem (Balakrishnan, 1978), we have demonstrated that the long-range correlations of the fractional Gaussian noise give rise to a power-law memory term. This term approximates the Caputo fractional derivative, as illustrated by Fig. S11(c), resulting in the fractional Langevin equation.

Third, to verify the assumptions and result of our analysis, we have experimentally validated the ability of the derived fractional Langevin equation to accurately explain the anomalous diffusive dynamics of large learning-rate GD. This is indicated by the strong agreement of theoretical and empirical TAMSDs (Fig. S11(d)).

In summary, this new analysis theoretically demonstrates that (discrete) gradient descent with a large learning rate approximates the (continuous) fractional Langevin equation. Taken together, along with the changes in response to the previous comment, we have demonstrated that our fractional diffusion theory is consistent with large learning-rate GD training in practice. To comprehensively detail this analysis, we have added Sec. 8 (L402, P. 15 to L544, P. 21) in the Supplementary Information (which replaces previous discussion of the overdamped FLE from the perspective of the Kac-Zwanzig model) and the new figure (Fig. S11). We have also rewritten a paragraph in the main text (L337, P. 12) to summarize the main ideas underlying this analysis. Based on this new analysis, we have accordingly updated equations in the main text as well as the simulation parameters stated in the Methods.

Fig. S11. Overdamped FLE accurately models GD dynamics with large learning rates on our minimal landscape model. **a**, The loss function approaches our landscape model as the number of scales, n , approaches infinity. **b**, The empirical variance of this loss function ($n = 10^4$) across 100 realizations agrees with our landscape model. **c**, The fractional Gaussian noise, which arises under the assumption of large learning rates, exhibits long-range correlations that induce a discrete memory kernel. This discrete memory kernel (solid lines) can be approximated by the continuous power-law kernel of the Caputo fractional derivative (dashed lines), resulting in the overdamped fractional Langevin equation. **d**, The overdamped fractional Langevin equation accurately predicts the anomalous diffusive dynamics of large learning-rate GD on a 1-D landscape ($H = 0.6$) as well as a 2-D landscape ($H = 0.6$).

AUTHOR RESPONSE

Reviewer #3

I would like to thank the authors for the reply and revision. This paper studies the neural network landscape and training in a novel perspective but the previous version failed to build clear connections to real applications. The advantages and regimes of the theory in this paper are now much clearer, especially in its relationship to practical settings. My concerns are now fully addressed, and I would recommend acceptance.

We thank the reviewer for taking the time to review our manuscript. We greatly appreciate the constructive comments, which have been very helpful in improving our paper.